# Cardiac radiotherapy induces electrical conduction reprogramming in the absence of transmural fibrosis

David M. Zhang [1,2], Rachita Navara [1,2], Tiankai Yin [2], Jeffrey Szymanski[3], Uri Goldsztejn [2,4], Camryn Kenkel[2,4], Adam Lang[5], Cedric Mpoy[3], Catherine E. Lipovsky[2,6], Yun Qiao[2,4], Stephanie Hicks[2], Gang Li[2,4], Kaitlin M. S. Moore[1,2], Carmen Bergom [1,3], Buck E. Rogers[3], Clifford G. Robinson[1,2,3], Phillip S. Cuculich[1,2,3], Julie K. Schwarz [1,3] & Stacey L. Rentschler [1,2,4,6✉]

Cardiac radiotherapy (RT) may be effective in treating heart failure (HF) patients with refractory ventricular tachycardia (VT). The previously proposed mechanism of radiation-induced fibrosis does not explain the rapidity and magnitude with which VT reduction occurs clinically. Here, we demonstrate in hearts from RT patients that radiation does not achieve transmural fibrosis within the timeframe of VT reduction. Electrophysiologic assessment of irradiated murine hearts reveals a persistent supraphysiologic electrical phenotype, mediated by increases in Na$_V$1.5 and Cx43. By sequencing and transgenic approaches, we identify Notch signaling as a mechanistic contributor to Na$_V$1.5 upregulation after RT. Clinically, RT was associated with increased Na$_V$1.5 expression in 1 of 1 explanted heart. On electro-cardiogram (ECG), post-RT QRS durations were shortened in 13 of 19 patients and length-ened in 5 patients. Collectively, this study provides evidence for radiation-induced reprogramming of cardiac conduction as a potential treatment strategy for arrhythmia management in VT patients.

[1] Center for Noninvasive Cardiac Radioablation, Washington University in St. Louis, School of Medicine, Saint Louis, MO, USA. [2] Department of Medicine, Cardiovascular Division, Washington University in St. Louis, School of Medicine, Saint Louis, MO, USA. [3] Department of Radiation Oncology, Washington University in St. Louis, School of Medicine, Saint Louis, MO, USA. [4] Department of Biomedical Engineering, Washington University in St. Louis, School of Medicine, Saint Louis, MO, USA. [5] Department of Pathology, Washington University in St. Louis, School of Medicine, Saint Louis, MO, USA. [6] Department of Developmental Biology, Washington University in St. Louis, School of Medicine, Saint Louis, MO, USA. ✉email: Stacey.Rentschler@wustl.edu

Sudden cardiac arrest, occurring in the setting of impaired cardiac function and structural heart disease that lead to ventricular tachycardia (VT) and ventricular fibrillation, is a leading cause of death in the world[1]. Implantable cardioverter defibrillators improve survival in at-risk populations at the cost of reduced quality of life from shocks and potential for worsening heart failure (HF). Anti-arrhythmic drugs (AADs) have considerable side effects and modest efficacy. Invasive radiofrequency catheter ablation (CA) identifies electrically reentrant regions within scar and delivers ablative thermal energy to homogenize scar and eliminate VT[2]. However, CA is time-consuming, technically challenging, and highly operator-dependent. Rates of acute VT cessation and durable control are high in favorable individuals, especially first-time CA patients with preserved ventricular function and ischemic cardiomyopathy with focal scar[2]. Success rates are considerably lower in patients with severely impaired ventricular function, diffuse scar, or prior CA[3–5]. Failure of CA may occur in part due to anatomical factors or limitations in the properties of heat transfer physics to create full-thickness, gap-free ablations[3–5].

Radiotherapy (RT) noninvasively delivers high-dose radiation to volumes of target tissue anywhere in the body, with minimal off-target exposure, and is well established for treating malignancies. Hypothetically, ablative doses of radiation could noninvasively replicate effects of CA, with a fibrotic response expected over months to years[6–8]. Recent clinical evidence has demonstrated the safety and efficacy of cardiac RT for treatment of refractory VT in patients[9–11]. The first case series reported a 99.9% reduction in VT burden after treatment with 25 Gray (Gy) ionizing radiation[9]. More recently, a prospective Phase I/II trial of 19 patients who had either previously failed or were not eligible for CA reported a 94% reduction in episodes of VT or premature ventricular contractions and a 12-month survival rate of 72% in this very sick population[10]. Similar outcomes have since been replicated by several independent medical centers[10–12]. In all cases, the therapeutic effect was observed within days to weeks, well before RT-induced fibrosis is expected to occur. As such, the mechanisms by which radiation reduces VT remain unknown.

Early preclinical studies that investigated the therapeutic potential for cardiac RT attempted to create transmural fibrosis[13–15]. In these models, higher doses between 40 and 160 Gy were sufficient to induce late-stage cardiac fibrosis and cause subacute atrioventricular block[16–22], while lower doses did not achieve these effects[7,16,17]. Importantly, these prior studies have not explained how doses in the range of 25 Gy can reduce VT burden acutely.

The presence of structural or electrical heterogeneity in the heart increases the probability of unidirectional block and reentrant tachycardia[23,24]. For sustained reentry, the conduction wavelength, defined as distance traveled by an electrical impulse while the myocardium is refractory, must be shorter than the path of conduction[24]. This wavelength is equal to the mathematical product of the effective refractory period (ERP) and conduction velocity (CV). Hence, the presence of long, tortuous substrates, fast repolarization, and slowed CV all predispose to reentry. CA prevents reentry by terminating conduction substrates, while management with Class III AADs prolongs the action potential duration (APD) and ERP to increase wavelength[25]. To date, there are no approved VT therapies that act through increasing CV. Though radiation is presumed to prevent arrhythmia by creating fibrosis in arrhythmogenic scar, the antiarrhythmic effects preventing reentry could potentially act through effects on ERP and/or CV.

In this study, we evaluated effects of ionizing radiation on the heart. Using explanted specimens from cardiac RT patients, we show that 25 Gy photon radiation, used in all clinical studies to date, was not sufficient to achieve CA-like scar. In mice and patients, we find that 25 Gy persistently increases levels of cardiac conduction proteins and enhances ventricular conduction. Through gain- and loss-of-function approaches, we identify cardiomyocyte Notch signaling as a potential mechanism whereby radiation may reprogram conduction. We also report a statistically nonsignificant difference in QRS intervals, with a tendency toward QRS shortening, in 19 patients treated with RT. Here we further discuss therapeutic strategies for reprogramming ventricular cardiomyocytes to a pro-conduction phenotype to prevent electrical reentry.

## Results

**Twenty-five Gy single-fraction radiation does not increase cardiac fibrosis in patients, despite marked decreases in VT burden.** We first investigated the association between radiation and cardiac fibrosis in 4 VT patients who received cardiac RT and provided postmortem cardiac specimens (Supplementary Table 1). Specimens targeted with 25 Gy radiation were stained with Masson's trichrome to visualize fibrosis, and targeted regions were compared with nontargeted remote specimens within the same hearts (Fig. 1a, b and Supplementary Fig. 1). Across samples, remote specimens received <5 Gy exposure. Patient A previously underwent (unsuccessful) radiofrequency CA that did not control subsequent VT episodes, and this catheter-ablated myocardium was used for comparison (Fig. 1c).

At baseline, individual hearts exhibited distinct levels of underlying fibrosis. Given that anatomic scarring is important for prescribing treatment locations and volumes[9], greater fibrosis is expected in radiation-targeted regions even prior to treatment. Using image-segmented analysis, we observed only minor differences in fibrosis between matched targeted and nontargeted regions of patients' hearts, and fibrosis was lower in all radiation-targeted myocardium when compared to CA myocardium (Fig. 1d). Independent of fibrosis, all four patients experienced substantial reductions in VT episodes in the 6 months following RT (Fig. 1e). Representative contrast-enhanced magnetic resonance imaging (MRI) scans of Patient E revealed no change in gadolinium enhancement and preserved myocardial tissue between baseline and at 3-month follow-up in the RT-targeted region (Fig. 1f), and there was no evidence of increased fibrosis on MRI post-RT in any patient. These clinical findings are consistent with previous preclinical studies that required doses in excess of 40 Gy to produce scar[16–22]. Collectively, these data strongly suggest that fibrosis alone cannot explain the clinical timeline and magnitude of reduced VT burden observed after RT.

**Twenty-five Gy radiation reprograms cardiac electrophysiology and increases conduction.** To further evaluate effects of 25 Gy irradiation (IR) to the heart, we assessed for structural and physiologic changes to the murine heart post-IR. To test whether post-mitotic mammalian cardiomyocytes recover from 25 Gy radiation-induced DNA damage, murine hearts were stained at 30 min, 3 h, and 24 h post-IR for phosphorylated histone H2AX (γH2AX), a marker of double-stranded DNA breaks[26]. Both percentages of γH2AX-positive nuclei and mean γH2AX intensity decreased with respect to time (Supplementary Fig. 2), a property of radioresistant tissues that demonstrates DNA repair and survival[27]. At 6 weeks post-IR, we performed Masson's trichrome staining to assess for gross evidence of fibrosis and did not detect significant collagen staining in either treatment condition after 6 weeks (Supplementary Fig. 3a). Furthermore, we observed no difference in collagen content between treated and untreated murine ventricles 6 weeks post-IR, as determined by quantitative hydroxyproline assay (Supplementary Fig. 3b).

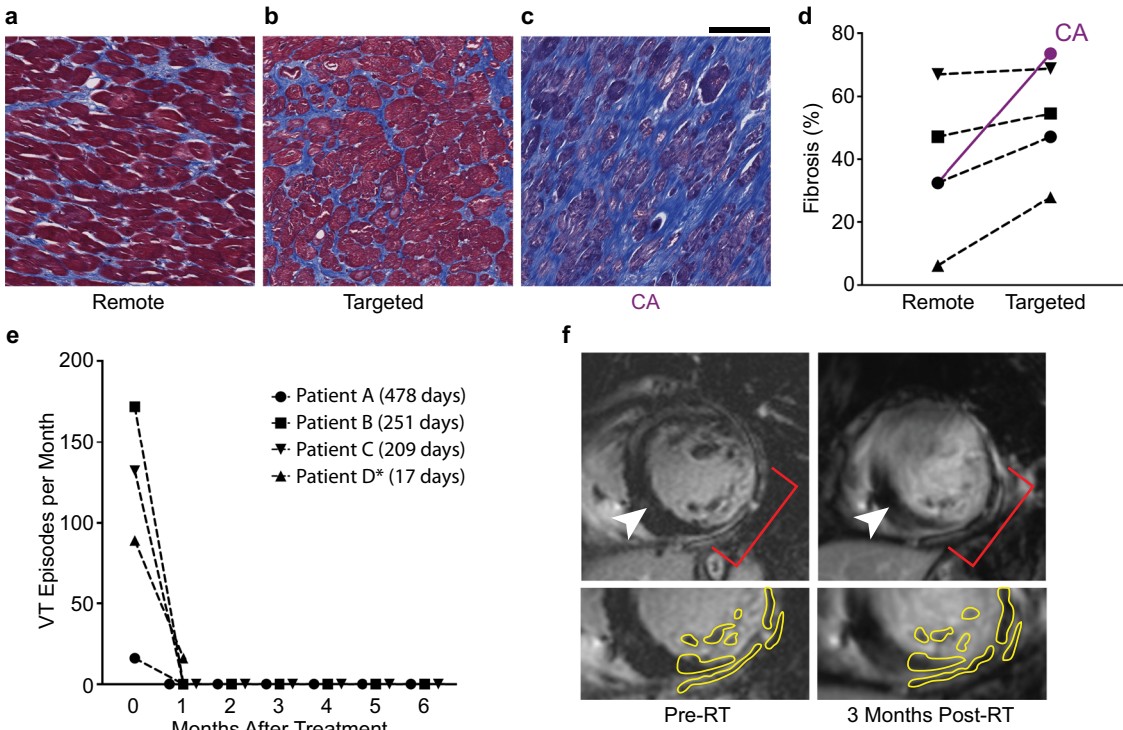

**Fig. 1 Cardiac fibrosis alone cannot account for the timing and effect of VT reduction after RT. a–c** Representative Masson's trichrome stains of regions of remote (**a**), targeted (**b**), and failed CA (**c**) myocardium in the same patient who received 25 Gy ionizing radiation for treatment of refractory VT. Scale bars = 500 μm. Experiment was independently repeated once on the same patient samples and produced similar results. **d** Percent fibrosis at the time of heart explant from 4 patients who received 25 Gy RT in control (left) and targeted (right) regions of myocardium. Percent fibrosis of the CA region for Patient A is shown in purple. **e** Defibrillator-recorded episodes of VT in the month before treatment and the 6 months after treatment for each patient. Days post-treatment until specimen collection are shown in the legend. *Patient D's values are divided by 20 to allow for comparisons on the same scale. **f** Gadolinium-enhanced MRI of a cardiac RT patient at baseline (left) and 3 months post-treatment (right). Top: the left ventricle with patchy, gadolinium-enhanced scar was transmurally targeted with 25 Gy between 3 and 6 O'clock (red brackets). Nonenhanced, remote myocardium is adjacent to target region (white arrowhead). Bottom: surviving nonenhanced myocardium within the same images is visible in the targeted region at baseline and 3 months post-treatment (yellow outline). Source data are provided as a Source data file.

Electrophysiologic and structural remodeling both contribute to arrhythmogenic substrates in the setting of disease. Because RT reduces VT in the absence of profound CA-like structural change, we evaluated electrophysiologic mechanisms by which IR may prevent reentry. We tested the effects of 25 Gy IR on adult murine cardiac electrophysiology at the clinically efficacious timepoint of 6 weeks post-IR, when a 99.9% reduction in VT burden was reported in patients[9]. While previous preclinical studies reported heart block in the 40–160 Gy dose range[7,16,19], 25 Gy radiation did not cause atrioventricular delay as measured by PR interval on electrocardiogram (ECG; Fig. 2a, b). However, a significantly shortened QRS interval was observed in IR mice compared to sham controls (Fig. 2a, c), while heart rate and other ECG intervals were unchanged (Supplementary Fig. 4a–c). To measure functional effects associated with QRS shortening, we optically mapped Langendorff-perfused mouse hearts. Ventricular CVs were significantly increased in IR mice compared to sham controls (Fig. 2d, e and Supplementary Fig. 4d), while APD at 80% repolarization ($APD_{80}$) and ERP were unchanged (Fig. 2f, g and Supplementary Fig. 4e). Together, these results suggest that 25 Gy radiation increases CV in the absence of profound changes to electrical repolarization, which favorably alters conditions against electrical reentry.

**Ionizing radiation increases cardiac conduction protein levels.** To understand how radiation increases CV, we quantified the major molecular determinants of cardiac conduction: cell size, the

cardiac sodium channel, and cardiac gap junctions. We detected no differences in cardiomyocyte cross-sectional area between the treatment groups (Supplementary Fig. 3c, d), and furthermore, the heart weight-to-tibia length ratio, a surrogate marker of hypertrophy, was unchanged (Supplementary Fig. 3e). Next, we performed western blots to assess for relative amounts of $Na_V1.5$, the pore-forming subunit of the cardiac voltage-gated sodium channel responsible for phase 0 depolarization. Densitometric quantification showed an approximately 80% increase in $Na_V1.5$ (Fig. 2h) within IR ventricles compared to controls. Immuno-blotting for connexin 43 (Cx43), the major subunit of the ventricular gap junction that allows for diffusion of charge, also demonstrated increased levels of Cx43 (Fig. 2i). Immunohis-tochemistry demonstrates that these proteins remain normally distributed in cardiomyocytes with no changes in channel localization between groups (Fig. 2j, k).

**Single fraction radiation durably reprograms the electrical substrate.** Patients receiving cardiac RT experience persistent reductions in VT burden[10,11,28]; therefore, we hypothesized that radiation-induced effects on cardiac conduction may persist over time. To better understand the chronic implications of single-fraction radiation for VT therapy, we asked whether CV, $Na_V1.5$ levels, and Cx43 levels remain persistently elevated. At 42 weeks, IR mice continued to express increased $Na_V1.5$ and Cx43 (Fig. 2l, m), and in this aged cohort, the QRS also remained shortened (Fig. 2n and Supplementary Fig. 5a–d). These persistent

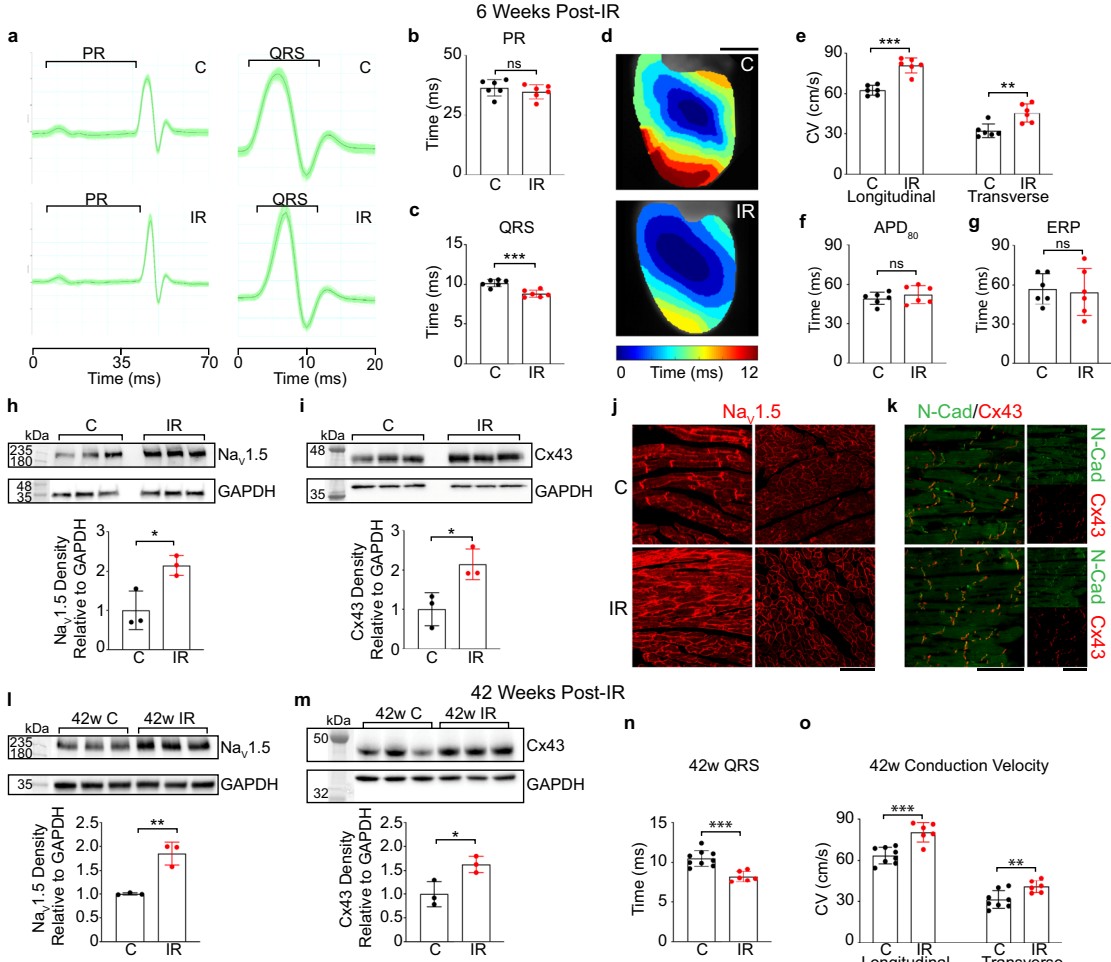

**Fig. 2 Cardiac RT persistently increases adult murine ventricular conduction. a** Representative ECGs from control (top) and IR mice (bottom) highlighting PR and QRS intervals. **b, c** Effect of radiation on PR ($P = 0.39$) and QRS (**$P = 0.00043$) intervals in control (black) versus IR (red) mice ($n = 6$ biologically independent animals per condition). **d** Representative ventricular activation maps from control and IR mice. Scale bars = 3 mm. **e** Effect of RT on ventricular CV ($n = 6$ biologically independent animals per condition; ***$P = 0.00005$; **$P = 0.0034$). **f, g** $APD_{80}$ ($P = 0.39$) and ERP ($P = 0.78$) in control versus IR mice ($n = 6$ biologically independent animals per condition). **h, i** Western blots of $Na_V1.5$ (*$P = 0.022$) and Cx43 (*$P = 0.025$) in control and IR ventricles ($n = 3$ biologically independent samples per condition). **j** Immunostaining of control (top) and IR (bottom) myocardium for $Na_V1.5$ and **k** co-stained for Cx43 (red) and N-Cadherin (green). Scale bars = 70 μm. Experiment was replicated three times in biologically independent specimens and produced similar results. **l, m** Western blots of $Na_V1.5$ (**$P = 0.0034$) and Cx43 (*$P = 0.027$) after 42 weeks post-IR ($n = 3$ biologically independent samples per condition). **n** Effect of RT on QRS intervals in control versus IR mice after 42 weeks (***$P = 0.00027$; C, $n = 9$; IR, $n = 6$). **o** Left ventricular CVs in control ($n = 9$ biologically independent animals) versus IR ($n = 6$ biologically independent animals) mice at 42 weeks post-IR (***$P = 0.00042$; **$P = 0.0092$). All $P$ values determined by two-way unpaired $t$ test. Bar graphs are represented as mean ± SD. Source data are provided as a Source data file.

molecular changes translated into persistently increased CVs (Fig. 2o and Supplementary Fig. 5e). Taken together, one single-fraction 25 Gy dose of IR is sufficient to achieve persistent electrical reprogramming of the heart, consistent with reports of patients with reduced VT burden 2 years after their RT treatment[10,28].

**Radiation dose-dependency of conduction reprogramming.** The current dose of cardiac RT that demonstrates robust clinical efficacy in humans is 25 Gy. This dose was selected based on data from preclinical studies that attempted to create radiation-induced fibrosis[19]. Given that the mechanism of RT could be mediated through electrophysiologic reprogramming, we asked whether doses <25 Gy are sufficient to achieve radiation-induced electrical changes to improve cardiac RT safety while maintaining efficacy. Using a dose de-escalation approach, we observed significantly faster CVs and shorter QRS durations in IR hearts treated with 20 and 15 Gy, while

doses of 10 or 5 Gy did not result in significant differences (Fig. 3a, b). As expected, ECG intervals were comparable among treatment groups prior to IR treatment (Supplementary Fig. 6), and we did not observe dose-dependent differences in other ECG intervals or heart rate at 6 weeks post-IR (Supplementary Fig. 7). Overall, the magnitude of conduction increase measured by CV and QRS tended to be greater in the 20 Gy group than in the 15 Gy group, though our study was not designed to detect differences between these two groups. Consistent with the functional dose response, western blotting of protein lysates revealed dose-dependent responses on expression of $Na_V1.5$ and Cx43 (Fig. 3c, d). Although the largest biological effects were seen at 25 Gy, these findings suggest that lower doses of ionizing radiation, between 15 and 20 Gy, may also be sufficient to achieve therapeutic electrophysiologic effects of RT on VT reduction.

**Radiation increases conduction within surviving myocardium in the setting of injury.** We tested effects of radiation on cardiac

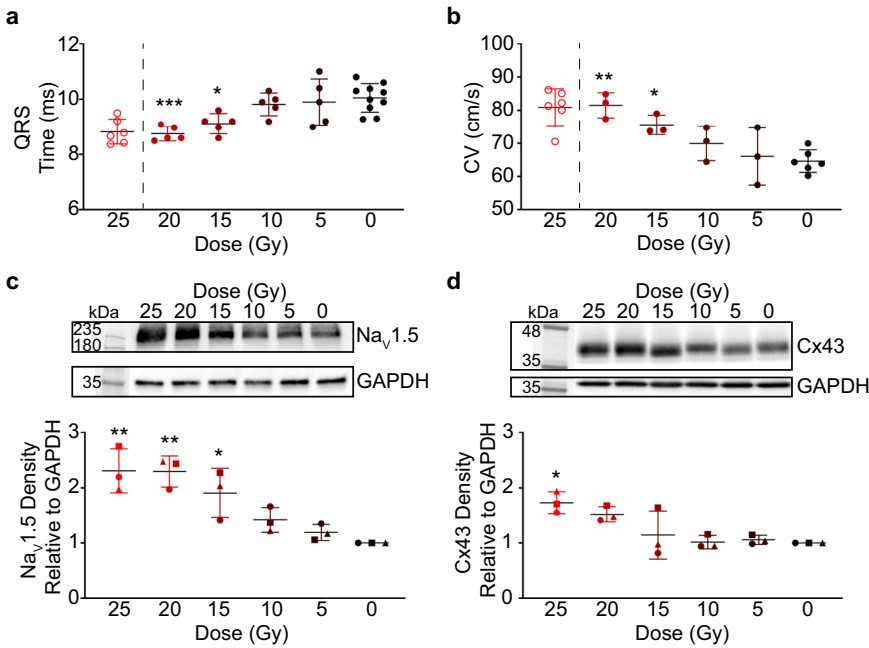

**Fig. 3 Conduction velocity reprogramming may be achieved at lower doses of radiation. a** Dose-dependent effects of radiation on QRS interval at 6 weeks post-IR ($n = 5$ biologically independent animals per IR condition; $n = 10$ biologically independent 0 Gy control animals; ***$P = 0.0009$, 20 versus 0 Gy; *$P = 0.023$, 15 versus 0 Gy). **b** Dose-dependent effects of radiation on longitudinal conduction velocity ($n = 3$ biologically independent animals per IR condition; $n = 6$ biologically independent 0 Gy control animals; **$P = 0.0023$, 20 versus 0 Gy; *$P = 0.046$, 15 versus 0 Gy). **c** Dose-dependent effects of radiation on Na$_V$1.5 protein levels by western blot ($n = 3$ biologically independent sample per condition over 3 independent experiments; **$P = 0.0014$, 25 versus 0 Gy; **$P = 0.0016$, 20 versus 0 Gy; *$P = 0.023$, 15 versus 0 Gy;). **d** Dose-dependent effects of radiation on Cx43 protein levels by western blot ($n = 3$ biologically independent per condition over 3 independent experiments; *$P = 0.011$, 15 versus 0 Gy). Values plotted from western blots were obtained from three independent western blots each denoted by a closed square, circle, and triangle. Twenty-five Gy QRS interval and CV values from Fig. 3 are plotted for reference (open circles) and are not used in statistical comparisons. Statistical analysis of all subpanels consisted of a one-way ANOVA followed by one-sided Tukey post hoc test of multiple comparisons. All bar graphs are represented as mean ± SD. Source data are provided as a Source data file.

conduction in the setting of injury and fibrosis, which better represents the myocardial substrate of the cardiac RT patient population. We utilized a surgical model of myocardial infarction (MI), in which the proximal left anterior descending (LAD) artery is permanently ligated, causing acute infarction of the apical left ventricle and subsequent replacement of apical myocardium by myofibroblasts and scar over a 2-week period[29]. Two weeks post-MI, mice received either 25 Gy radiation or sham treatments, followed by downstream optical mapping experiments at 6 weeks post-IR.

Consistent with electrically inert scar[30], direct stimulation of the scar region did not result in electrical capture or propagation of impulse in any heart (Fig. 4a, gray region). Border zone (BZ) myocardium above the LAD ligation was stimulated and CV through viable myocardium was quantified, including the anatomic BZ region defined as the 2 mm segment of tissue scar junction, but these impulses did not further propagate into the scar region (Fig. 4a, gray region, and Supplementary Fig. 8a). In conducting myocardium, 25 Gy radiation significantly increased both longitudinal and transverse CVs (Fig. 4a, b), which included isochrones in the anatomic BZ near the scar. Based on these observations, we postulated that increases to levels of Na$_V$1.5 and Cx43 occur primarily in viable cardiomyocytes and not myofibroblasts. To further delineate the regions exhibiting increased Na$_V$1.5 and Cx43 after IR, ventricles from MI and MI + IR mice were dissected into distinct regions of mid-basal BZ myocardium and apical scar. As expected, the presence of ion channel proteins was much higher in the conducting BZ myocardium than in the electrically inert scar region at baseline (Supplementary Fig. 8b, c). Western blotting of protein lysates from the conducting BZ myocardium revealed increased expression of Na$_V$1.5 and Cx43 in

the MI + IR hearts compared to MI controls (Fig. 4c, d), while these effects were not observed in regions of electrically inert scar tissue in the same hearts (Fig. 4e, f).

**Gene expression and Notch signaling define distinct post-IR states at 2 and 6 weeks**. To identify genes and pathways differentially expressed after IR, we profiled the ventricular transcriptome by bulk RNA sequencing (RNA-seq) in control mice and at 2 and 6 weeks post-IR. Overall, 509 genes were differentially expressed across sample groups (log fold change >1 and Benjamini–Hochberg adjusted significance of $P < 0.05$), and the greatest number of expression changes were observed comparing control and 6-week time points (Supplementary Fig. 8a, b). In Gene Set Enrichment Analysis using Molecular Signatures Database (MSigDB) hallmark gene sets, canonical stress response pathways were elevated at both timepoints (Fig. 5a). We observed that several of the most differentially expressed genes were implicated in stress responses to radiation (Supplementary Table 2), and many have known pro- or anti-fibrotic functions in the heart (Supplementary Fig. 9c, d).

Within the most upregulated response pathways, we also explored signaling responses previously implicated in the modulation of cardiac conduction. The Notch signaling pathway, which plays an important role in conduction system development[31–33], was present at both timepoints (Fig. 5a). Furthermore, leading edge analysis revealed Notch pathway-associated genes (*Cdkn1a*, *Jag1*, *Jag2*, and *Notch2*) driving enrichment of multiple pathways. Given that Notch signaling is quiescent in resting adult cardiomyocytes, we investigated in detail the differential expression of all genes associated with the

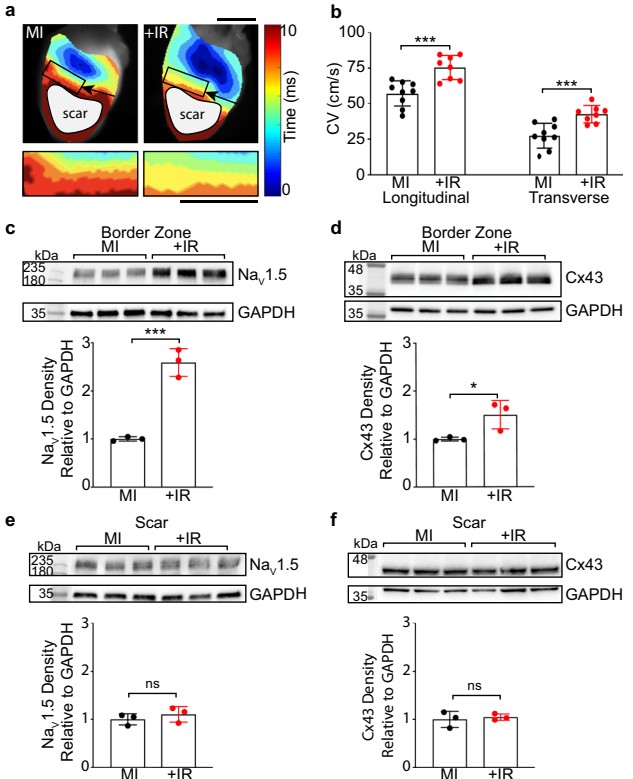

**Fig. 4 Radiation specifically reprograms the border zone myocardium in a murine model of myocardial infarction. a** Top: representative ventricular activation maps from control myocardial infarction (MI, left) and irradiated MI (MI+IR, right) mice at 8 weeks post-MI and 6 weeks post-IR. Black arrows point to location of left anterior descending (LAD) artery ligation. Black box denotes approximate 2 mm border zone at the scar/myocardium interface. Regions of scar that were epicardially stimulated but did not capture are denoted by gray circles. Magnified isochrones in the border zone region used to calculate transverse conduction velocities. Scale bars = 5 mm. **b** Left ventricular conduction velocities in MI (black) versus MI+IR (red) hearts (MI $n = 9$ biologically independent animals; MI+IR $n = 8$ biologically independent animals; longitudinal ***$P = 0.0006$; transverse ***$P = 0.0009$;). **c, d** Western blot (top) and quantified density (bottom) of $Na_V1.5$ (***$P = 0.0007$) and Cx43 (*$P = 0.041$) protein in MI and MI+IR border zone myocardium superior to the LAD ligation ($n = 3$ biologically independent samples per condition). **e, f** Western blot (top) and quantified density (bottom) of $Na_V1.5$ ($P = 0.42$) and Cx43 ($P = 0.68$) protein in MI control and MI+IR apical scars ($n = 3$ biologically independent samples per condition). All $P$ values determined by two-way unpaired $t$ test. All bar graphs are represented as mean ± SD. Source data are provided as a Source data file.

murine Notch pathway gene ontology (GO) term. Unsupervised hierarchical clustering of Notch GO genes grouped samples into their respective timepoints, and each pairwise comparison showed significant changes in Notch genes, with a trend toward Notch activation (Fig. 5b). Given this broad signal, we also investigated gene expression within Notch pathway functional classes, using genes with known relevance to cardiomyocytes, which further suggested cardiac Notch activation (Fig. 5c) and pointed to the potential role for IR-induced Notch reactivation in the heart.

**Transient Notch activation persistently reprograms the ventricular electrical substrate**. Previously, we reported that

transient cardiomyocyte Notch reactivation resulted in persistent epigenetic and electrophysiologic changes that produced a conduction-like phenotype[33,34]. In these studies, Notch reactivation upregulated *Scn5a*, which encodes for $Na_V1.5$, and increased sodium current ($I_{Na}$)-mediated cardiomyocyte membrane excitability[34]. To further determine whether cardiomyocyte Notch signaling can increase cardiac conduction, we utilized the Tet-on transgenic system to transiently and selectively activate Notch in adult cardiomyocytes (inducible Notch intracellular domain (iNICD); *αMHC-rtTA*; *tetO_NICD*)[34,35]. Adult iNICD mice were fed doxycycline chow for 3 weeks to transiently activate Notch, followed by a minimum of 16 weeks washout for iNICD turnover. This doxycycline pulse was chosen to sufficiently activate Notch while mimicking a transient dose of RT-induced activation. When compared to littermate controls, longitudinal CVs of iNICD ventricles were significantly increased after 16-week washout (Fig. 6a, b). In a separate cohort, mice were sacrificed after a 52-week washout to test whether Notch activation leads to persistent upregulation of cardiac conduction proteins. Western blotting revealed that transient activation of Notch signaling was associated with persistent increases in ventricular $Na_V1.5$ (Fig. 6c), but not Cx43 (Fig. 6d). These findings suggest that transient Notch activation may partially contribute to radiation-induced persistent upregulation of the cardiac sodium channel and are important to understanding a mechanism whereby a single dose of radiation could lead to permanent changes in the electrical substrate.

**Loss of cardiomyocyte Notch signaling partially rescues radiation-induced conduction reprogramming**. To test whether cardiomyocyte Notch signaling is not only sufficient but also necessary for reprogramming conduction after radiation, we utilized cardiomyocyte-specific, inducible Notch loss-of-function transgenic mice (Notch iLOF, *αMHC-MerCreMer* and *R26r^dnMAML/+*). Notch iLOF mice express a tamoxifen-inducible Cre recombinase under the cardiomyocyte-specific myosin heavy chain promoter, as well as a loxP-flanked STOP cassette upstream a dominant-negative mutant of the mastermind-like Notch transcriptional coactivator (*dnMAML*), which represses nuclear transactivation of the NICD[36]. Upon tamoxifen-induced Cre recombination, Notch iLOF mice permanently express *dnMAML* in cardiomyocytes to effectively and specifically knock out Notch signaling in a cell-autonomous manner[34]. Littermates expressing *αMHC-MerCreMer* and receiving tamoxifen but lacking the *dnMAML* mutant were used as controls.

Following tamoxifen-induced Cre recombination and washout, Notch iLOF mice and littermate controls were treated with 25 Gy radiation. In non-irradiated sham mice, there was no baseline difference in CV between treatment groups (Supplementary Fig. 10a). Although radiation was associated with increases in CV, $Na_V1.5$, and Cx43 in both groups (Supplementary Fig. 10b–d), Notch iLOF attenuated the post-IR conduction increase by >30%, resulting in significantly decreased CVs in Notch iLOF mice compared to littermate controls (Fig. 7a, b and Supplementary Fig. 10e). Following radiation, Notch iLOF mice expressed significantly decreased $Na_V1.5$ expression (Fig. 7c), while we detected no difference in Cx43 expression (Fig. 7d and Supplementary Fig. 10f). From this observed partial rescue, these data suggest that radiation-induced electrical reprogramming is partially regulated by Notch-induced upregulation of the cardiac sodium channel.

**Clinical cardiac RT is associated with molecular and functional changes in electrophysiology**. To translate our preclinical findings into patients receiving RT for treatment of VT, we asked

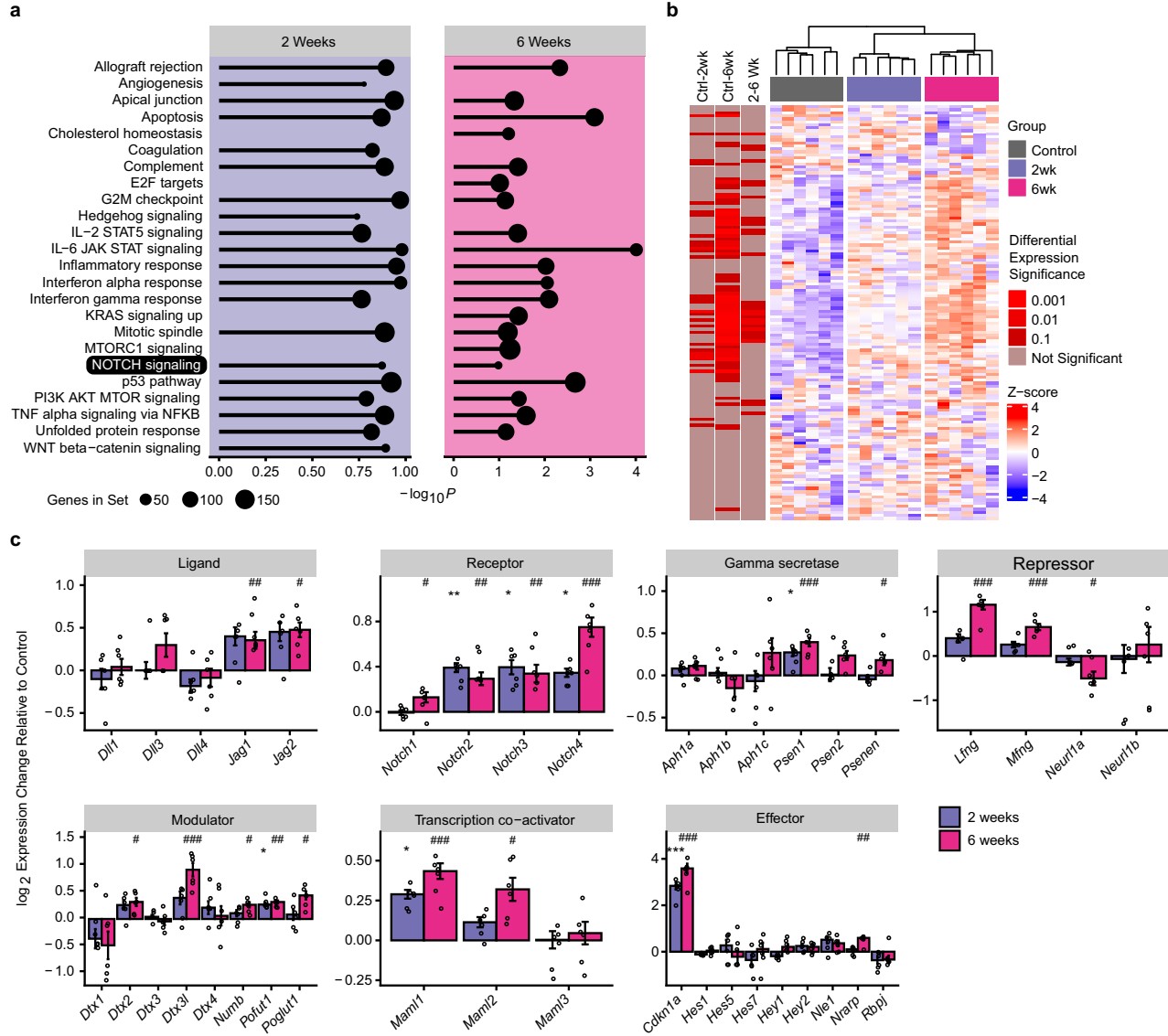

**Fig. 5 Post-IR states exhibit distinct transcriptional signatures and activation of the Notch signaling pathway. a** Top 20 transcriptionally over-represented hallmark pathways at 2 and 6 weeks post-IR. The Notch signaling pathway is present at both timepoints, and Notch pathway-associated genes *Cdkn1a*, *Jag1*, *Jag2*, and *Notch2* were drivers of enrichment in other over-represented pathways. **b** Heatmap of Notch GO term genes. Samples group by hierarchical clustering of Notch GO gene expression. **c** Cardiac-related Notch pathway members grouped by function. Bar graphs represent log₂-fold change at 2 weeks or 6 weeks post-IR relative to sham control state and error bars represent standard error ($n = 6$ biologically independent samples per treatment group, *indicates comparisons of 2 weeks post-IR versus control, #indicates comparisons of 6 weeks post-IR versus control, */#$P < 0.10$, **/##$P < 0.01$, ***/###$P < 0.001$). *Jag1* ##$P = 0.0049$; *Jag2* #$P = 0.03$; *Notch1* #$P = 0.076$; *Notch2* **$P = 0.0066$, ##$P = 0.0012$; *Notch3* *$P = 0.014$, ##$P = 0.0019$; *Notch4* *$P = 0.015$, ###$P = 7.2E{-}6$; *Psen1* *$P = 0.051$, ###$P = 0.00078$; *Psenen* #$P = 0.024$; *Lfng* ###$P = 1.0E{-}6$; *Mfng* ###$P = 0.00023$; *Neurl1a* #$P = 0.092$; *Dtx2* #$P = 0.051$; *Dtx3l* ###$P = 0.00014$; *Numb* #$P = 0.061$; *Pofut1* *$P = 0.019$, ##$P = 0.0049$; *Poglut1* #$P = 0.020$ *Maml1* *$P = 0.016$, ###$P = 0.00031$; *Maml2* #$P = 0.010$; *Cdkn1a* ***$P = 0.00033$, ###$P = 2.5E-6$; *Nrarp* ##$P = 0.0019$. Adjusted *P* value statistics determined by Benjamini–Hochberg procedure applied to two-sided limma differential expression values to control false discovery from multiple comparisons ("Methods"). Source data are provided as a Source data file.

whether radiation is sufficient to upregulate Na$_V$1.5 and/or Cx43 in an explanted heart previously targeted with 25 Gy, compared to a nontargeted remote region within the same failing heart. At the time of Patient F's heart transplant due to refractory non-ischemic HF (965 days post-RT), the superior portion of the basal left ventricle, previously targeted with 25 Gy (Segment 1), was collected for flash freezing, and the inferior-mid left ventricle (Segment 10), which received <5 Gy radiation exposure (Supplementary Fig. 11), was used as a nontargeted control (Fig. 8a). To control for potential baseline regional differences in protein

expression, matched tissue from segments 1 and 10 were collected from 2 non-failing donor hearts that were rejected for transplantation. Overall, the level of Na$_V$1.5 in the nontargeted region (Segment 10) of the nonischemic failing heart was much lower than in analogous segments of nonfailing donor hearts (Fig. 8b), consistent with known reductions in Na$_V$1.5 in the setting of HF[37–40]. After 965 days post-RT, we observed threefold higher levels of Na$_V$1.5 within the targeted myocardium when compared to the remote region in Patient F's heart, restoring Na$_V$1.5 in the targeted region to levels analogous to non-failing ventricles

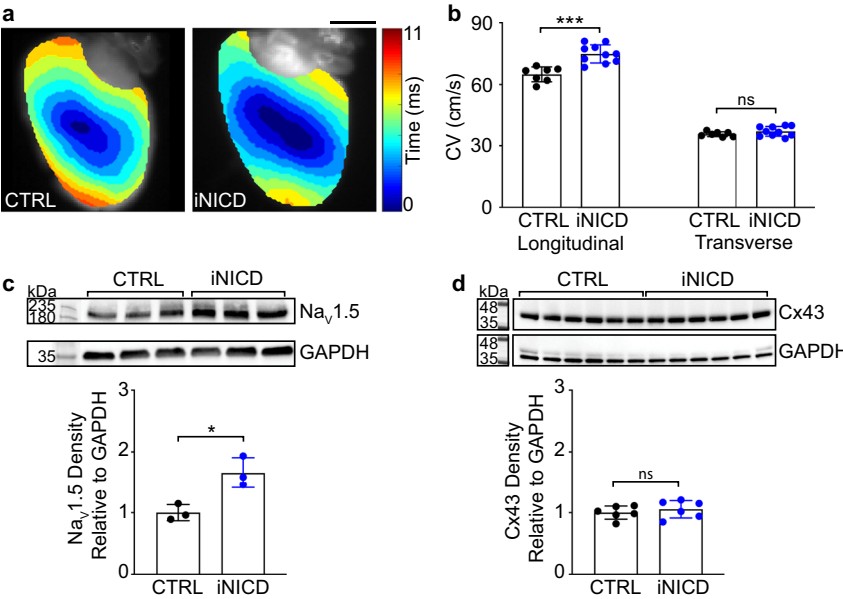

**Fig. 6 Transient iNICD-mediated, cardiomyocyte-specific Notch activation persistently increases conduction velocity and Na$_V$1.5 protein expression. a** Representative activation maps and **b** conduction velocity measurements of control (CTRL, black) *αMHC-rtTA; tetO_NICD* (iNICD, blue) murine hearts after 3-week doxycycline induction and 16-week washout ($n = 7$ biologically independent CTRL mice; $n = 9$ biologically independent iNICD mice; Longitudinal ***$P = 0.00019$, Transverse $P = 0.18$). **c, d** Western blot and quantified densitometry from CTRL versus iNICD hearts after 3-week doxycycline induction, 52-week washout for Na$_V$1.5 ($n = 3$ biologically independent samples per condition, *$P = 0.026$) and Cx43 ($n = 6$ biologically independent samples per condition, $P = 0.47$, ladder was run in parallel on a separate gel to allow for quantitative comparisons on the same blot). All $P$ values determined by two-way unpaired $t$ test. All bar graphs are represented as mean ± SD. Source data are provided as a Source data file.

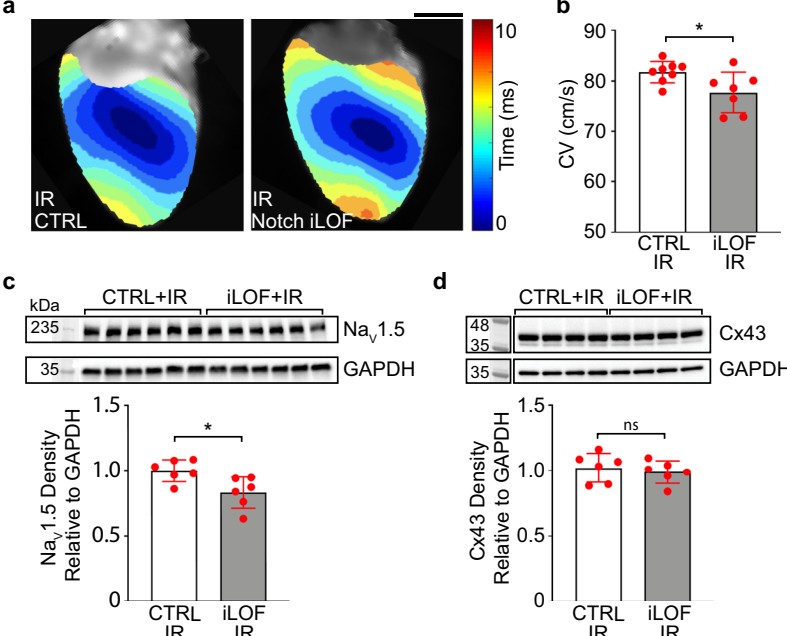

**Fig. 7 Loss of cardiomyocyte Notch signaling inhibits radiation-induced conduction reprogramming. a** Representative activation maps and **b** longitudinal conduction velocity measurements of irradiated littermate control (left, CTRL, white bars) and *αMHC-MerCreMer; R26r$^{dnMAML}$/+* (right, iLOF, gray bars) murine hearts at 6 weeks post-IR. $n = 8$ biologically independent CTRL mice; $n = 7$ biologically independent iLOF mice; *$P = 0.027$. **c, d** Western blot and quantified densitometry of Na$_V$1.5 ($n = 6$ biologically independent samples per condition, *$P = 0.012$) and Cx43 ($n = 6$ biologically independent samples per condition across two blots run in parallel, $P = 0.59$,). All $P$ values determined by two-way unpaired $t$ test. All bar graphs are represented as mean ± SD. Source data are provided as a Source data file.

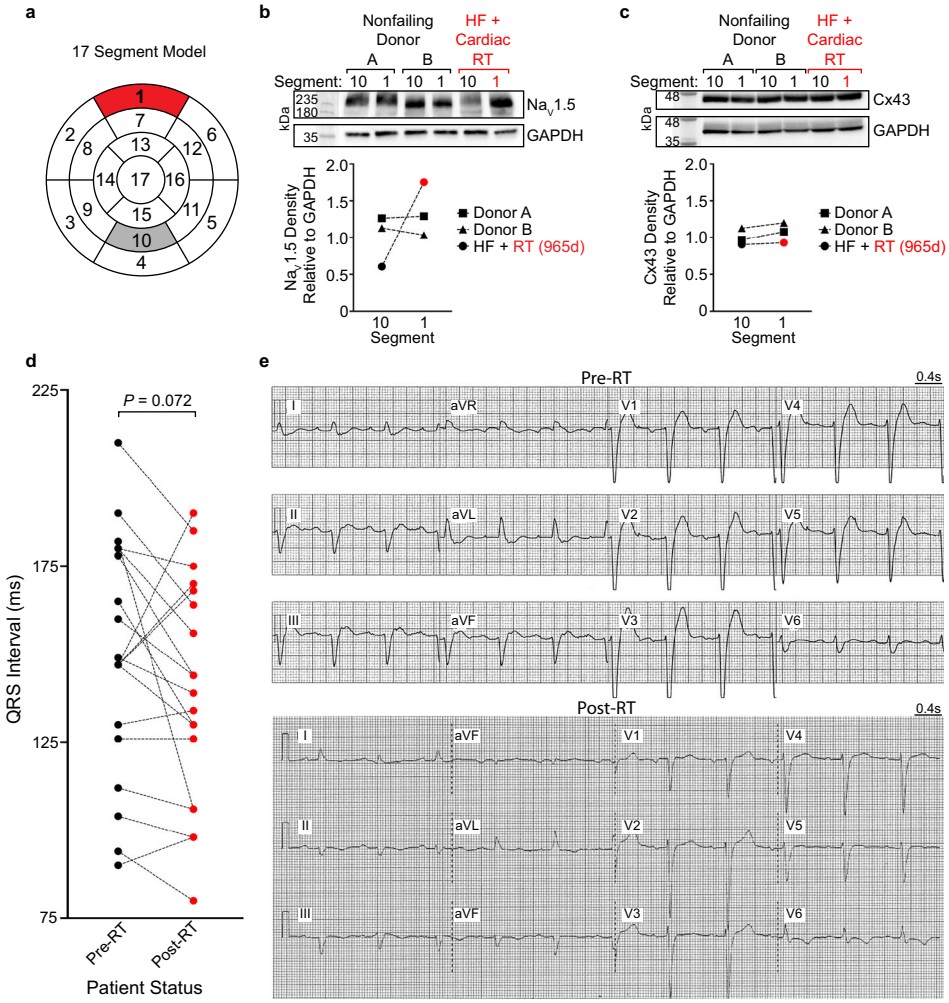

**Fig. 8 Reprogramming in VT patients treated with 25 Gy RT. a** Cardiac 17-segment model illustrating 25 Gy targeted (Segment 1, red) and remote nontargeted (Segment 10, gray) myocardium collected from a heart failure patient with nonischemic cardiomyopathy and VT (Patient F), who previously received cardiac RT. The nontargeted Segment 10 received <5 Gy radiation exposure. For comparison, corresponding Segments 1 and 10 were collected from 2 nonfailing, non-RT donor hearts rejected for transplantation. **b** Western blot and quantified densitometry of Na$_V$1.5 in Segment 10 versus Segment 1 of nonfailing (Donors A and B) and RT-treated (HF+Cardiac RT, red) hearts. **c** Western blot and quantified densitometry of Cx43 in Segment 10 versus Segment 1 of nonfailing and RT-treated hearts. **d** Matched QRS durations in the ENCORE-VT patient cohort at baseline (pre-RT, 149 ± 34 ms) versus 6-week post-RT (139 ± 32 ms) timepoints (n = 19 paired individual patients, P = 0.072 by two-tailed Wilcoxon signed-rank test). **e** Representative 12-lead ECG of robust QRS shortening recorded from Patient G at the time of RT treatment, with a baseline QRS of 165 ms and presence of LBBB, and then after 6 months post-RT, exhibiting shortened QRS (130 ms) and resolution of LBBB. Examples of pre- and post-RT ECGs of the remaining three patients that exhibited robust QRS shortening in the ENCORE-VT patient cohort, as well as an example of an unchanged QRS interval, are provided in the Supplementary Figures. Source data are provided as a Source data file.

(Fig. 8b). In this nonischemic cardiomyopathy, there was no observed decrease in the levels of Cx43 at baseline, and likewise no observed difference in Cx43 between the targeted and non-targeted segments (Fig. 8c). Differential expression between Na$_V$1.5 and Cx43 within these regions could potentially be explained by heterogeneous gap junction remodeling in non-ischemic HF when compared to ischemic HF[41].

Next, we analyzed serial surface ECGs of patients who received cardiac RT. In contrast to the mouse, where the whole heart is irradiated, only the arrhythmogenic region of a patient's heart is targeted. Thus, not all RT patients would be expected to exhibit a shortened QRS duration. Even still, we observed a statistically nonsignificant difference in QRS durations within the 19-patient cohort by 6 weeks post-RT (149 ± 34 ms pre-RT versus 139 ± 32 ms post-RT, n = 19, P = 0.072, Fig. 8d). Overall, compared to baseline intervals, post-RT QRS durations were

shorter in 13 patients and longer in 5 patients (Fig. 8d). We also present examples of robust RT-associated QRS shortening, defined strictly as a decrease in the QRS duration by ≥25 ms on serial ECGs, in at least 4 out of the 19 patients (Fig. 8e and Supplementary Figs. 12–14). As an example, Patient G's QRS shortened from 165 ms with left bundle branch block (LBBB) to a QRS interval of 130 ms without LBBB, and this QRS remained shorted for at least 6 months (Fig. 8e). Presented in the Supplementary Figures are the pre- and post-RT ECGs of the remaining 3 patients who exhibited persistent QRS shortening (Supplementary Figs. 12–14), as well as an example where there was minor to no shortening observed (Supplementary Fig. 15). Consistent with our preclinical data in mice, these ECG changes may be potential examples of radiation-induced functional electrical reprogramming in humans.

## Discussion

Cardiac RT is an emerging treatment option for patients with refractory VT. The objective of this study was to define the structural, functional, and molecular effects of focused, single-fraction RT in the adult mammalian heart. Our findings suggest that radiation may be used therapeutically to modulate cardiac electrophysiology for VT management.

Through histologic examination and MRI scans of the hearts from patients who received cardiac RT, we provide evidence from a cohort of human RT patients to show that 25 Gy transmural radiation does not create transmural scar. These findings are consistent with preclinical studies that required doses in excess of 40 Gy to produce lesions that interrupted conduction[17,22]. Importantly, emerging preclinical and clinical reports continue to cite fibrosis and scar homogenization as the preferred biologic effect after RT and often discuss whether 25 Gy treatments result in underdosing to the target. Yet, clinical VT reductions occur within weeks following 25 Gy RT, and radiation-induced fibrosis alone cannot account for the magnitude and timing of these effects. Indeed, our study further contributes to the growing body evidence to suggest that, for cardiac RT, dose strategies may not necessarily be chosen based on the anticipated effects of late-stage, radiation-induced fibrosis to achieve therapeutic effect.

Previous preclinical studies have reported increases in global levels and lateralization of Cx43 after carbon radiation to the heart[42–44], and increased gap junction formation was reported after radiation to other tissue types[45,46]. In these studies, the mechanism of Cx43 upregulation and the functional consequences of lateralization remain unclear. In the current study using photon-based IR, we also observed IR-associated increases in Cx43 protein; however, we did not observe changes in Cx43 lateralization or conduction anisotropy. Future studies are warranted to determine whether there are differences in $Na_V1.5$ and Cx43 responses, as well as dosing considerations, among different radiation modalities, including X-ray, carbon ion, and proton beam cardiac RT.

Reduced sodium channel expression and availability frequently occur in the setting of heart disease or injury and are major causes of acquired arrhythmias[37–40]. Slowed conduction due to reduced sodium current density is a critical factor in sustaining reentry in regions of fibrosis[47,48]. Previous large animal studies have demonstrated feasibility of gene therapy delivery of skeletal muscle sodium channel SkM1[49,50] and connexins[51] to increase cardiac conduction and prevent VT. Here we observe acute, persistent increases in electrical propagation within IR-treated hearts mediated by increases in $Na_V1.5$ and Cx43. Indeed, increases in either sodium channel expression or gap junction coupling are expected to prevent reentrant arrhythmias[47–51] via restoration of cardiac conduction[52,53].

Previously, reactivation of developmental signaling pathways, including Notch, was sufficient to epigenetically reprogram cardiomyocytes to produce a conductive "Purkinje-like" electrophenotype[33,34]. Furthermore, Notch signaling is known to play a role in the radiation response and radio-resistance of several tissue types[54,55]. Through RNA-seq and transgenic approaches, we identify cardiomyocyte-specific Notch activation as one potential cell signaling mechanism that may contribute to the pro-conductive effect of radiation via upregulation of the cardiac sodium channel. In this study, we utilized murine model systems to allow for genetic manipulation and testing of direct causality of radiation-induced conduction reprogramming. Specifically, we demonstrate that cardiomyocyte Notch activation is both sufficient to increase cardiac conduction and necessary to achieve the full effects of radiation-induced conduction reprogramming through gain- and loss-of-function approaches.

The functional effects of increased $Na_V1.5$ and Cx43 may act through several electrical mechanisms to prevent conduction block and reentry and explain the antiarrhythmic effect conferred to patients after 25 Gy RT. In the setting of heart disease and conduction heterogeneity, homogenization of conduction through restoration of either membrane excitability or inter-cellular coupling would reduce the vulnerable window that allows unidirectional block to occur[56]. Additionally, increased inter-cellular coupling alone may prevent aberrant conduction by increasing the functional sink and reducing the safety factor to terminate arrhythmic conduction[53]. However, this phenomenon requires Cx43 upregulation that exceeds the effect size observed our study. Finally, data from our current study suggests that, within surviving cardiomyocytes in the peri-infarct region, upregulation of $Na_V1.5$ and Cx43 results in local increases in CV to functionally rescue regions of conduction delay necessary for sustained reentry and tachycardia[53]. Future studies are warranted to determine which molecular effects can be best leveraged in the setting of RT to prevent reentry and tachycardia.

Our study presents findings to suggest that an antiarrhythmic therapy, successfully used in patients with refractory VT, may increase levels of the cardiac sodium channel and improve conduction. Indeed, explanted cardiac specimens obtained from a treated patient with refractory VT revealed threefold higher $Na_V1.5$ protein levels in the radiation-targeted region when compared to a nontargeted region of the same heart, restoring $Na_V1.5$ to levels within the range of nonfailing myocardium. As an observation of the potential effect of RT on human physiology, we also report a nonsignificant decrease in mean QRS duration in the Electrophysiology-guided Noninvasive Cardiac Radioablation for Ventricular Tachycardia (ENCORE-VT) patient cohort, as well as robustly shortened QRS intervals in at least 4 of the 19 patients.

Our findings have direct relevance for patient care. Most surviving patients continue to exhibit reduced VT burden 24 months after a single RT treatment[28]. Our results demonstrate that the functional and molecular effects of RT and Notch reactivation are persistent and expected to directly translate into long-term durability of therapy. A dosage of 25 Gy was prescribed in previous clinical studies to minimize cardiotoxic effects of radiation. Although this current dose is proven to be safe and effective[10], there are rare cases of pericardial effusion and a single adverse event of gastropericardial fistula that occurred 2 years after treatment[28]. Here we show that radiation-induced electrical reprogramming can also occur at lower doses in the range of 15–25 Gy in mice. Indeed, Notch signaling has been reported to activate in a variety of tissue types following lower doses of radiation[54,55,57]. These data could warrant future clinical studies of careful downward dose titration to find the lowest effective dose that minimizes risk of late-stage toxicity. Lower dosages may be particularly favorable in instances when the prescribed treatment volume is large or when the prescribed treatment location is adjacent to nearby radiosensitive organs.

There are several limitations to our present study. Most patients who received cardiac RT survive with reduced VT burden for years after treatment. Because of the durability of this emerging therapy, available postmortem samples remain limited, and results consequently remain difficult to generalize. This limitation further emphasizes the need for preclinical cardiac RT studies. Given that each patient is prescribed a target volume and location specific to the individual's scar-related VT, we also acknowledge that global electrophysiologic changes reflected by changes to the QRS interval is neither observed nor expected in every patient. Although we observed 13 patients with shortened QRS durations after RT, these QRS changes were statistically

nonsignificant in our 19-patient cohort and at least 5 patients exhibited lengthened QRS durations. Overall, these changes in QRS duration cannot be excluded as an unrelated phenomenon in the absence of additional analyses, and a comprehensive analysis of all ECG parameters is outside of the current scope of our study. This study's clinical results are also limited by the trial's single-arm design, low number of patients enrolled, narrow patient selection, and recruitment and treatment conducted at a single academic center, which currently limit generalization of results to other patient populations. In our preclinical study, we profiled radiation response after whole-heart radiation, while, clinically, RT is limited to select arrhythmogenic regions. Indeed, future studies are expected to elucidate relationships among treatment target volume, isodoses, local changes in conduction protein expression, and global changes in ECG intervals. Additionally, we implemented a surgical infarct model of HF, which does not represent the subset of patients with nonischemic cardiomyopathy. Furthermore, while humans and large animals have exhibited HF-associated decreases in conduction proteins, rodent models of HF have smaller conduction substrates, exhibit remodeling of repolarizing currents[58], and thus make reentrant tachycardia difficult to demonstrate. Future studies in large animal models of VT are warranted. We also acknowledge that there are important differences between the ionic currents that make up the action potentials of human and murine cardiomyocytes. While excitatory sodium current $I_{Na}$ is well conserved among mammals[59], there are several differences in the expression of repolarizing potassium currents across species. For example, the rapidly ($I_{Kr}$) and slowly ($I_{Ks}$) activating delayed rectifier potassium channels responsible for repolarization of the human cardiomyocyte action potential do not exist in mice[59]. Although we do not observe effects of IR on ERP, our study does not rule out an effect of radiation on human cardiomyocyte repolarization. Finally, we highlight cardiomyocyte Notch signaling as a mechanism by which $Na_V1.5$ is upregulated after radiation, while acknowledging that the mechanisms of post-RT reprogramming are likely multifactorial. Future studies will undoubtedly further define the cellular mechanisms underlying these effects.

In summary, this study supports a model whereby a single-fraction dose of 25 Gy radiation persistently restores electrical propagation in diseased hearts without causing transmural fibrosis and scar. Mechanistic insights such as these are expected to lead to refinements in the cardiac radiation clinical protocol and ultimately greater adoption of cardiac RT. The ability to noninvasively and functionally eliminate VT circuits, by modulating cardiac electrophysiology in a localized way and without tissue destruction, may allow for safer treatments and prevention of arrhythmias for individuals with myocardial scarring.

## Methods

**Study design**. This study aimed to determine the structural and electrophysiologic effects of single-fraction ionizing radiation on the adult mammalian heart. To this end, we utilized several experimental approaches. In fibrosis studies, we characterized levels of fibrosis in irradiated human and murine myocardium. In electrophysiologic studies, we studied the effects and mechanisms of cardiac ventricular conduction increase following 25 Gy ionizing radiation in healthy and diseased murine myocardium and following doxycycline-induced activation of Notch signaling and tamoxifen-induced loss of Notch signaling in cardiomyocytes. We also show examples of sodium channel upregulation in one of one and potential QRS duration changes in VT patients treated with 25 Gy RT.

In patients, total sample size was powered to demonstrate acute safety and preliminary efficacy of noninvasive radiation therapy for treatment of refractory ventricular tachycardia under the ENCORE-VT (NCT02919618) clinical trial, a single-arm Phase I/II trial, in which 19 patients were enrolled and treated[10]. Patients with refractory VT (i.e., failed at least one CA or was contraindicated to CA and failed/became intolerant to at least one antiarrhythmic medication) and met all inclusion/exclusion criteria were enrolled into the study. Patient-level data were restricted to paired, binary intra-patient comparisons (non-targeted versus targeted and/or pretreatment versus posttreatment).

For studies in mice, power analyses were used to determine minimum group sizes a priori. These power calculations was performed using G*power (v.3.1.9). The sample size calculation for each cohort was based on a *t* test to observe changes in CV or ERP. A study with an effect size of 0.25 and a power of 80% required a minimum of $n = 6$ per group to test this difference at 5% significance using a two-tailed test. Mice were randomized to treatment condition and both male and female littermates were used for all experiments. For experiments that used aged mice or when surgery was performed, we initially overpowered these cohorts in anticipation of procedure-related death such that there would be adequate power at the experimental endpoint. For western blot on samples obtained from wild-type and iNICD mice, a study with an effect size of 0.80 and power of 80% was initially chosen and required a minimum of $n = 3$ independent biological replicates per group to test this difference at 5% significance using a two-tailed test. In anticipation of a smaller effect size in Notch LOF rescue experiments, a study was designed to test for an effect size of 0.25 and power of 80%, and these also required a minimum of $n = 6$ independent biological replicates per group. This sample size was also subsequently used to power a test for Cx43 expression in iNICD mice when effect size appeared lower than the $Na_V1.5$ effect size. For RNA-seq, an $n = 6$ biologically independent samples per group was chosen to detect meaningful transcriptional changes in heart tissue based on previous publications[60].

**Human specimen and data collection**. This study was approved by the Washington University School of Medicine Institutional Review Board (IRB). Patient samples and data were collected under the ENCORE-VT clinical trial. Only patients who provided informed consent prior to specimen collection were included, and all human specimen experiments were performed in accordance with IRB-approved analyses. At the time of heart transplant or death (autopsy), whole-heart specimens were obtained from patients who previously underwent non-invasive cardiac radiation for treatment of refractory VT and consented to specimen collection. Each specimen was compared with its corresponding radiation treatment plan to identify regions of the heart that were either treated with 25 Gy (targeted) or anatomically distant from radiation (remote). Across all samples, remote regions of myocardium received radiation exposure that was <5 Gy. In specimens obtained from autopsy, targeted and remote regions of patient hearts were resected and fixed in 10% formalin upon collection for histopathology. In one explanted heart collected from transplant, targeted and nontargeted tissue were flash frozen immediately at the time of explant for molecular experiments and protein measurements. Remaining specimens obtained from autopsy and formalin fixed were neither flash frozen nor collected for protein measurements, as these specimens were obtained days after death, during which substantial protein degradation was expected, and were only appropriate for clinical histopathology. Cardiac MRIs were taken according to the ENCORE-VT clinical trial guidelines. Two patients were treated as part of our institution's previously published case series[9] prior to establishment of the ENCORE-VT clinical trial and for whom MRI was not taken. While quantification of fibrosis using gadolinium enhancement was not feasible due to the presence of imaging artifacts, we include a representative set of images from a single patient that illustrates the lack of any observed change in fibrosis or cardiac injury in our patient cohort.

**Patient ECG interpretation**. Pre- and Post-RT patient ECGs were taken according to the ENCORE-VT clinical trial (NCT02919618). QRS intervals were measured independently by two blinded investigators, and values obtained were subsequently compared and verified against one another. While some patients exhibited smaller changes in pre- versus post-RT QRS intervals, robust QRS shortening was strictly defined as a decrease in the QRS interval by at least 25 ms on serial surface ECGs.

**Human specimen fibrosis quantification**. Trichrome-stained fixed human specimens were scanned into high-resolution SVS image files using an Aperio ImageScope (v.12.1 Leica Biosystems). SVS image files contain seven magnification levels stored as Gaussian pyramids. All fibrosis quantification was performed using MATLAB (2017b, MathWorks) through a four-step process of sampling, preprocessing, quantization, and calculation[61]. In the sampling step, each SVS file was parsed into 900 equally spaced 50 px-by-50 px images that span the entire high-resolution file in order to allow for parallelization of analysis and prevent out-of-memory errors. Next, preprocessing was performed using simple linear iterative clustering with a compression ratio of 1:50, to generate superpixels while preserving tissue boundaries[62]. Image files were subsequently converted from the RGB Colorspace to the L*a*b* Colorspace for image representation that is independent of brightness and illumination, which may vary across samples that were subject to differences in sample thickness, microscope settings, and batch-dependent differences during staining. Quantization subsequently defined pixel parameters, which corresponded to areas of collagen, red blood cells, myocytes, or background, as defined by $K$-means[63]. Parameters were chosen based on the results obtained from four training samples and kept constant for all computations. All pixels were then assigned to their corresponding category using Lloyd–Max algorithms[64]. Overall, percent fibrosis was defined by the relationship $\text{fibrosis} = \frac{\text{Area}_{\text{collagen}}}{\text{Area}_{\text{collagen}} + \text{Area}_{\text{RBC}} + \text{Area}_{\text{myocytes}}}$. For fibrosis analyses, the mean computation time was approximately 50 min per sample.

**Mouse studies.** Adult male and female littermate outbred wild-type CD-1 mice were purchased (Charles River Laboratories, Wilmington MA). iNICD (αMHC-rtTA and tetO_NICD)[33,35] and Notch iLOF (αMHC-MerCreMer and R26r[dnMAML/+])[34] mice were maintained on a mixed genetic background. iNICD and Notch iLOF mice were genotyped for rtTA/tetO_NICD and Cre/Rosa26r, respectively (Supplementary Table 3). In all experiments, age- and gender-matched littermate control animals were used for comparison. In all IR experiments, CD-1 mice aged ≥8 weeks were used. For experiments involving conditional iNICD expression, doxycycline chow was fed (200 mg/kg, BioServ). Adult iNICD mice were fed doxycycline chow for 3 weeks to transiently activate Notch, followed by a minimum of 16 weeks regular chow to allow for sufficient doxycycline washout and iNICD turnover. In iLOF experiments involving tamoxifen-induced Cre recombination, mice were fed tamoxifen chow for 10 days to allow for Cre recombination, followed by a 4-week washout prior to radiation or sham treatment.

All mice were housed in 12-h light/dark cycles, at ambient temperatures of 20–22 degrees Celsius, at a humidity range between 40 and 60%, and with access to food and water ad libitum, in accordance with animal study guidelines at Washington University. Animal protocols were approved by the Animal Studies Committee at Washington University School of Medicine, and all animals were handled in accordance with the National Institutes of Health Guide for the Care and Use of Laboratory Animals.

**Murine cardiac IR model.** Murine IR treatment was achieved using the small animal radiation research platform (SARRP, Xstrahl Inc.). The SARRP has an X-ray tube mounted on a gantry, with a motorized stage serving as a couch on which the mouse is placed. These allow for cone-beam computed tomography (CT) imaging (imaging mode) and radiation delivery (therapy mode)[65]. In imaging mode, the SARRP operates at 60–80 kVp and 0.5 mA using 1 mm of aluminum filtration. In therapy mode, the SARRP operates at 175–220 kVp with 0.15 mm of copper filtration. This platform allows for image-guided isocentric radiation of murine hearts and mimics RT therapy in humans.

Mice were first anesthetized using isoflurane and then cone-beam CT imaging was performed to target the heart. The CT images were reconstructed and imported into Muriplan (v.3.0. Xstrahl Inc.), where the isocenter was selected. The heart was then irradiated using anterior–posterior-opposed beams from the 10 mm × 10 mm collimator at a 3.9 Gy/min dose rate. A single-fraction exposure to the entire volume of the mouse heart was delivered at the predetermined dose. For comparisons, all littermate (sham) controls received isoflurane anesthesia and CT imaging, but no isocentric ionizing radiation.

In short-term, nonsurgical murine experiments, one irradiated Notch iLOF mouse spontaneously died at 4 weeks post-IR, while all remaining sham control and IR mice survived to their experimental endpoints. In the aged, long-term murine IR cohort, 2 sham control mice (each at 35 weeks and 42 weeks after sham treatment) and 4 IR mice (each at 18 weeks, 19 weeks, 31 weeks, and 41 weeks post-IR) spontaneously died after treatment. On necropsy, there was no evidence of direct cardiac damage or gross structural differences in either treatment group. Of note, 3 of the 4 cadaveric IR mice appeared notably cachexic compared to surviving littermates.

**Murine MI model.** Adult mice were anesthetized with ketamine (100 mg/kg) and xylazine (10 mg/kg) administered intraperitoneally. Under artificial ventilation, a left thoracotomy was performed to expose the left ventricle and LAD artery[29]. The proximal portion of the LAD was ligated with a 9-0 silk suture. After closure of the surgical incision, mice were recovered on a warmer and returned to their cage. The mean time for each surgical procedure was approximately 10 min. From 29 mice surviving the initial surgery, 8 died before the end of the 2-week postoperative period, and surviving mice were randomly assigned to receive either 25 Gy IR or sham treatment. During the 6-week period following radiation treatment, one sham mouse and zero IR mice died. At the time of optical mapping (8 weeks post-MI, 6 weeks post-IR), hearts from three mice (one control and two IR) were covered in excess surgical adhesions and/or adhered to the anterior thoracic wall and these hearts could not be dissected without causing ventricular wall rupture and were excluded from downstream optical mapping experiments. In both treatment groups, stimulation of myocardium outside the scar region resulted in optical action potentials within the scar region that had <50% upstroke velocity and peak optical voltage when compared to regions of viable myocardium (Supplementary Fig. 8a), and thus isochrones below the surgical ligation are omitted.

**Murine electrocardiogram.** Surface ECGs were performed on mice anesthetized with isoflurane. Lead II ECGs were recorded at a sampling rate of 4 kHz using LabChart (Version 8.0, ADInstruments), and ECG intervals of 100 averaged beats were measured using the LabChart ECG software package[32]. Because the morphology of the QRS complex in mice includes the presence of a J wave that immediately follows the S wave, the end of the QRS interval was strictly defined as the time when the S wave crosses the isoelectric line. During all ECG collection and analysis, investigators were blinded to sample group allocation.

**Optical mapping.** Murine hearts were explanted, Langendorff-perfused, and optically mapped, as previously described[60,66]. Specifically, hearts were perfused

and immersed in an oxygenated Tyrode's solution (128.2 mM NaCl, 4.7 mM KCl, 11.1 mM glucose, 1.19 mM NaH$_2$PO$_4$, 20 mM NaHCO$_3$, 1.05 mM MgCl$_2$, 1.3 mM CaCl, pH 7.4) maintained at 37 °C. Briefly, the potentiometric dye Di-4-ANEPPS (Life Technologies) and the excitation–contraction uncoupler blebbistatin (Cayman Chemical) were added to the Langendorff perfusate to allow for fluorescent detection of membrane potential changes and to eliminate motion artifacts, respectively. Murine hearts were electrically stimulated at 10 Hz to measure CV or with S1–S2 stimulation protocols of decreasing cycle lengths to determine ERP. A green 524 nm light source was used to excite the voltage-sensitive dye. The emitted fluorescence was filtered by a single long-pass 650 nm filter and recorded by CMOS camera (BrainVision LLC). Data were collected through the MiCAM Ultima software (v.2011.11, BrainVision LLC). Data analysis was performed on MATLAB using the open source Rhythm software (Rhythm2014b, http://efimovlab.org/research/resources/rhythm)[66]. With surface coordinate distances and the time of action potential upstroke, defined as maximum voltage derivative $(dV_m/dt)_{max}$ of optical signals, activation maps were generated for calculations of longitudinal and transverse CVs. Velocities were calculated by using isochrone distances and times of action potential upstroke originating from the site of stimulation. All investigators were blinded to the sample group allocation during data collection and analysis.

**Histology and immunohistochemistry.** Immunohistochemistry was performed on OCT-embedded frozen or paraffin-embedded sections. Gross heart morphology and collagen content were examined using Masson's trichrome stain (American MasterTech Scientific). Sections were treated with antibodies recognizing γH2AX (R&D Systems AF2288, lot KNH1218111), Na$_V$1.5 (1:100, Alomone Labs, 493-511, lot ASC005AN3002 and ASC005AN3702), Cx43 (1:50, Thermo Fisher 71-0700, lot PI209083), and N-Cadherin (1:100, Invitrogen 33-3900, lot UC284646). Secondary fluorescent-conjugated antibodies included Alexa 568 (1:200, Abcam ab175471, lot 1494753) and Alexa 488 (1:200, Abcam ab150077, lot 3244688, and ab150121, lot 3201847). Sections were also treated with wheat germ agglutinin (Thermo Fisher) to measure cardiomyocyte size and 4′,6-diamidino-2-phenylindole (Sigma D9542) to identify nuclei. For all digital imaging, investigators were blinded to sample conditions, and for each immunohistochemical target, all samples were imaged with identical settings on a confocal microscope (Leica Biosystems). Quantification of blinded images was performed using ImageJ (v.2.0.0, NIH).

**Hydroxyproline assay.** A hydroxyproline assay was performed using a standard 4-hydroxyproline amino acid colorimetric test kit (Sigma). Briefly, 10 mg of flash-frozen tissue was homogenized in water and hydrolyzed at 120 °C for 3 h[35]. After hydrolysis, oxidized hydroxyproline was reacted with 4-(dimethyamino)benzaldehyde to produce a colorimetric product proportional to hydroxyproline concentration. Hydroxyproline content for standard curves and samples was quantified by measuring absorbance at 560 nm using the Gen5 software (v.2.0.4, Agilent BioTek) and Excel (v.16.52, Microsoft).

**Protein extraction, electrophoresis, and immunoblotting.** Protein lysates were made from flash-frozen mouse or human ventricular samples for immunoblotting[60]. For each biological replicate, approximately 20–30 mg of ventricular tissue was homogenized using a modified RIPA lysis buffer (50 mM Tris-HCl, 150 mM NaCl, 5 mM EDTA, 1% Triton-X, 0.5% sodium deoxycholate, 0.1% sodium dodecyl sulfate) treated with a cOmplete Mini EDTA-free protease inhibitor tablet (1 tab/10 mL buffer, Roche). After homogenization, samples were centrifuged for 10 min at 13,200 × g, and supernatants were aliquoted and stored at −80 °C. Protein concentrations were measured using a commercial BCA assay (Thermo Fisher). Gel electrophoresis was performed using pre-cast 4–15 or 4–20% Tris-Glycine gels (BioRad) and pre-cast 10 or 12% NuPage Bis-Tris gels (Invitrogen). After gel electrophoresis, proteins were transferred overnight to polyvinylidene difluoride (Millipore Sigma) or nitrocellulose (Invitrogen) membranes. Blots were blocked for 1 h in milk blocking buffer (5% milk in Tris-buffered saline, 0.05% Tween-20). Immunoblotting was performed using anti-Na$_V$1.5 (1:1000 Alomone Labs, 493-511, lot ASC005AN3002 and ASC005AN3702), anti-Cx43 (1:2000 Invitrogen 71-0700, lot PI209083), and anti-GAPDH (1:2000, Cell Signaling Technology 14C10, lot 14) and incubated overnight at 4 °C. Secondary detection was performed using horseradish peroxidase-conjugated antibodies (1:5000 Abcam ab6721, lot GR3321356) and Clarity Western ECL Substrate (BioRad). Densitometry analysis was performed using the ImageLab software (BioRad) and normalized to GAPDH density. For all immunoblotting experiments, blots were repeated with a minimum of three technical triplicates using the same biological samples. For all densitometric quantifications, investigators were blinded to sample conditions.

**RNA isolation.** Total RNA was isolated from murine ventricles using TRIzol reagent (Invitrogen)[35], per the manufacturer instructions, and DNAse-treated using the TURBO DNA-free Kit (Ambion).

**RNA sequencing.** RNA-seq was performed on murine ventricular tissue from sham control mice ($n = 6$) at 2 weeks post-IR ($n = 6$) and at 6 weeks post-IR ($n = 6$). Total RNA was isolated as described above, and RNA was quantitated and

quality-assessed using a 2100 Bioanalyzer (Agilent). RNA-seq was performed by the Genome Technology Access Center at Washington University School of Medicine. Briefly, sequencing libraries were constructed using the Roche KAPA RNA HyperPrep Kit per the manufacturer's recommendations. Libraries were sequenced with the Illumina NovaSeq S4 ($2 \times 150$ bp) to a median depth of 42.7 million $+/-$ 3.8 million reads.

Reads were aligned to the *Mus musculus* GRCm38.76 genome using STAR (v. 2.7.3a.)[67], and alignment quality metrics were obtained from RNA-SeQC (v. 2.3.5). Transcript abundance was quantified in Salmon (v. 1.0)[68], and downstream analysis was performed in R (v. 3.6.3). Transcript abundance was summarized to gene counts using tximport (v. 3.10). Gene counts were scaled to library size using trimmed mean of *M*-value normalization in EdgeR (v. 3.28)[69]. Genes with counts per million of <0.23 (10/median library size) in at least 6 samples were removed. Weighted likelihoods were calculated and differential expression analysis performed in limma (v. 3.42)[70]. Resulting *P* values were adjusted for multiple comparisons by Benjamini–Hochberg method. Pathway analysis and gene set enrichment analysis were performed in GSEA (v. 4.1.0)[71] for each pairwise comparison using 1000 permutations and otherwise default parameters. Annotated gene sets tested were from the MSigDB using Broad Institute-hosted mouse annotations[72].

**Statistical analysis**. Prism version 8.02 (GraphPad Software Inc.) was used to perform statistical analyses and plot data. Distribution normality was tested using D'Agostino–Pearson test. Unless otherwise stated, a $P < 0.05$ (95% confidence) was considered statistically significant. For comparisons of two experimental groups, a two-tailed, unpaired Student's $t$ test was used. For comparisons of three or more experimental groups, a repeated-measures one-way analysis of equal variance with Tukey post hoc test of means was used. For statistical comparisons for differences between pre-RT and post-RT QRS intervals in ENCORE-VT patients, a two-tailed Wilcoxon matched-pairs signed-rank test was performed. For all data representation, the P value, number of biological replicates, and statistical test used are reported in corresponding figure legend.

**Reporting summary**. Further information on research design is available in the Nature Research Reporting Summary linked to this article.

## Data availability
All the data supporting the findings from this study are available within the manuscript and its Supplementary Information. Transcriptomic data that support the findings of this study have been deposited with GEO under accession number GSE153981. Annotated gene sets were obtained from the Molecular Signatures Database (MSigDB) (https://www.gsea-msigdb.org/gsea/msigdb/genesets.jsp?collection=H). Source data are provided with this paper.

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

## Acknowledgements

This work was supported by NIH grant numbers T32 HL134635 (D.M.Z.), T32 GM07200 (D.M.Z.), R01 HL130212 (S.L.R.), UH3 HL141800 (S.L.R.), and S10 OD020136. This study received seed funding from the Department of Radiation Oncology, Cancer Biology Division, at Washington University (J.K.S. and S.L.R.). Dr. Schwarz holds a Female Investigator Award from the AACR-Bristol Meyers Squibb and funding from the Radiological Society of North America. Dr. Rentschler holds a Career Award for Medical Scientists from the Burroughs Wellcome Fund and funding from The Foundation for Barnes-Jewish Hospital that directly supported this work. The authors would like to thank Dr. Yoram Rudy for providing valuable feedback during the study. We thank Carla Weinheimer at the Washington University Mouse Cardiovascular Phenotyping Core for performing surgical ligation procedures. We also thank the Genome Technology Access Center in the Department of Genetics at Washington University School of Medicine for RNA sequencing and assistance in genomic analysis.

## Author contributions

D.M.Z., U.G., J.K.S., and S.L.R. conceived the study, designed experiments, and participated in data interpretation. D.M.Z., R.N., T.Y., U.G., C.K., C.E.L., Y.Q., S.H., and G.L. performed experiments and analyzed experimental data. J.S. performed bioinformatic analyses and assisted in designing the experiments. A.L. collected and interpreted pathology. U.G. performed image-segmented pathology analysis. C.M., C.B., B.E.R., and J.K.S. developed treatment plans for animal experiments. K.M.S.M. coordinated the ENCORE-VT study and collected clinical data and specimens. R.N. and P.S.C. interpreted ECGs. C.G.R. and P.S.C. developed patient treatment plans, collected and analyzed patient data, and participated in data interpretation. C.B., C.G.R., P.S.C., and J.K.S. provided scientific feedback. D.M.Z. and S.L.R. drafted the manuscript. All authors read, contributed to editing, and approved the manuscript.

## Competing interests

The authors declare the following competing interests: C.G.R. and P.S.C. have filed two institution-owned patents: WO2017078757 (Noninvasive Imaging and Treatment System for Cardiac Arrhythmias) that relates to overall methods for delivery of cardiac radiation in patients and WO2019118640 (System and Method for Determining Segments for Ablation) that relates to the use of cardiac segments for cardiac radiation targeting. C.G.R. and P.S.C. also provide consulting services to Varian, which produces linear accelerators for radiation treatment delivery. C.G.R. and P.S.C. declare no other competing interests. No other authors declare any competing interests.
