## [Peer Review File · Nature Communications]

REVIEWER COMMENTS

Reviewer #1 (Remarks to the Author):

In this manuscript, Zhang et al. sought to investigate the mechanism of antiarrhythmic action of cardiac radio-ablation. Their main findings are that (i) radiation-induced fibrosis was mild at 25 Gy (clinical dose) and (ii) radiation induced a long-lasting upregulation in NaV1.5 and Cx43 expression. The authors conclude that radiation improved myocardial conduction, leading to reentrant ventricular arrhythmia suppression.

Cardiac radio-ablation is a novel treatment modality for patients with refractory ventricular arrhythmias and its mechanisms of action are unknown. The two leading hypotheses are somewhat contradictory: (i) radiation-induced dense fibrosis vs (ii) connexin upregulation and improved conduction. This manuscript provides compelling evidence supporting the latter.

Comments

1. The authors do not provide any data or discussion as to how radiation may lead to increased Nav1.5/Cx43; or how why these changes should persist over time.
2. The findings that radiation upregulates Cx43 expression and improves conduction is not entirely novel, as cited by the authors.
3. The antiarrhythmic effect of Na1.5/Cx43 upregulation is not demonstrated. The radiation-induced increase in conduction velocity may make the anatomic pathlength too short to sustain reentry, but no evidence is presented to this effect. The paper would be substantially strengthened if results in an animal model with inducible VT were presented.

4. Radiation did not result in substantial fibrosis in the animal model. Previous animal studies found 25 Gy to result in at least moderate fibrosis (ref 25, for example). What accounts for this difference?

5. The narrowing of QRS is presented for a single patient. The authors presumably have access to a much larger sample size. Presenting what happens to the QRS for the ENCORE-VT cohort would be more convincing.

6. The change in QRS duration observed in the animal models is relatively small, corresponding to relatively small changes in conduction that seem both inconsistent with the substantial changes in both Nav1.5 and Cx expression.

7. What is the difference between the data in Fig 3c and Fig 3n of the paper? The authors show differences between QRS duration of control and irradiated mouse hearts. What were the QRSs before irradiation? Were they comparable at that time point in the 2 groups?

8. Data should be shown to support the functional significance of the Nav1.5 upregulation in terms of Na current in irradiated vs control mouse cardiomyocytes.

9. The authors show evidence of accelerated conduction in the normal myocardium outside the scar zone, which they attribute to increased Na current and Cx function. What is the evidence that these effects can restore conduction in the peri-infarct regions of conduction block that lead to scar-related VT?

Reviewer #2 (Remarks to the Author):

In this report, Zhang and colleagues examine the effects of cardiac radioablation on electrophysiologic properties using various methods. The observation that has driven this study is an observed incongruity between predicted pro-fibrotic effects and the clinically observed antiarrhythmic effects of cardiac radioablation. The authors examine post-mortem/transplant specimens from patients who have undergone cardiac radioablation, and they studied a murine model of cardiac radioablation.

The key findings are: lack of significant fibrosis seen in the irradiated myocardium, and distinct changes in conduction properties, manifest as increased conduction velocity, with observed upregulation of key cellular conduction protein components, including Nav1.5 and CX43. Additional dose finding studies suggest the possibility that lower doses of radiation may provide similar electrophysiologic tissue effects.

These findings provide a plausible mechanism for clinical observations of relatively early antiarrhythmic effects. The authors are to be applauded for performing a comprehensive, multi-modality approach to studying this phenomenon.

There remain a few questions in this reviewer's mind that would benefit from a response:

- The challenge of the post-mortem/transplant portion of the study is the limited number of samples, which is understandable. In addition however, the internal "control" of non-irradiated remote portions of ventricular myocardium very likely received some radiation exposure. It would be important to specify, based on isodose curves, the estimated dose received by these "control" samples. If in fact these tissues received anything above 15Gy, as the authors themselves have noted, it is likely these areas should have been expected to exhibit similar electrophysiologic changes.
- How can localized effect on target result in global ECG changes? It is entirely understandable how in the whole-heart irradiated murine model global ECG changes are seen, as the entire heart is exposed to proposed threshold radiation dosing. However, Sub-25Gy dosing, and especially sub-15Gy dosing is presumably given to the rest of ventricle? How then can localized cellular EP changes result in global ECG changes as suggested by the authors in their sample patient's ECG? It would seem entirely plausible that this was an unrelated observation, unless there can be demonstration of global cellular changes to most of the ventricle due to a large mass of supra-threshold exposed myocardium.
- To further expand on the above two points, in a larger heart model, geographic sampling to correlate isodose delivery to cellular observations would be both enlightening and potentially clinically relevant in terms of expectations for treatment outcomes and cardiac effects.
- It would be very reasonable, and perhaps enlightening to spend a sentence or two discussing hypotheses in more detail for why increased CV may result in antiarrhythmic effects. Possible mechanisms may include reduction of conduction Safety Factor (see Shaw and Rudy – authors' institution!, 1985) from creation of source-sink mismatch, possibly through recruitment of surrounding myocardium. Alternatively, loss of a function zone of slow conduction may functionally eliminate the critical element needed to sustain reentry. Further discussion on this is certainly at least hypothesis generating.

- The dose finding studies are an interesting result – one concern is of course if loss of beneficial antiarrhythmic effect is observed. Would testing of varying doses then be warranted clinically?

- There is also a clinical observation in multiple centers' outcomes of late VT recurrence (6-12month range) in some patients. How would the authors hypothesize an explanation for these observations given the findings of this study?

Minor point:

- Line 58 “especially those with ??? ventricular function and ischemic cardiomyopathy with focal scar” – there is a missing word here - ?relatively preserved?

Reviewer #3 (Remarks to the Author):

This is a very thorough and well-written article with nice illustrations. I have few comments. Please address those pointwise.

Earlier studies have already shown that much higher dose of radiation (40-160 Gy) is needed to induce transmural fibrosis. Therefore, the first part of the study design (to show if 25 Gy causes transmural lesions) has no novelty. Please discuss.

Sample size is very small. Thus, it is difficult to generalize the findings, which is a major limitation of the paper.

Was MRI used in all to define myocardial fibrosis in all?

Was Bonferroni correction performed to calculate the significance for the 509 genes?

Please provide the baseline characteristics of the 4 patients including BMI and comorbidities.

Depending on their observation that conduction velocity increased with shorter QRS complex in hearts treated with 15 Gy, are the authors going to change the standard dose from 25 to 15 Gy in their clinical practice?

Reviewer #1 (Remarks to the Author):

In this manuscript, Zhang et al. sought to investigate the mechanism of antiarrhythmic action of cardiac radio-ablation. Their main findings are that (i) radiation-induced fibrosis was mild at 25 Gy (clinical dose) and (ii) radiation induced a long-lasting upregulation in Nav1.5 and Cx43 expression. The authors conclude that radiation improved myocardial conduction, leading to reentrant ventricular arrhythmia suppression.

Cardiac radio-ablation is a novel treatment modality for patients with refractory ventricular arrhythmias and its mechanisms of action are unknown. The two leading hypotheses are somewhat contradictory: (i) radiation-induced dense fibrosis vs (ii) connexin upregulation and improved conduction. This manuscript provides compelling evidence supporting the latter.

Thank you for this thoughtful assessment of our study. Please see our responses below.

Comments

1. The authors do not provide any data or discussion as to how radiation may lead to increased Nav1.5/Cx43; or how why these changes should persist over time.

Thank you for this feedback. We agree that more data and discussion to understand mechanisms of persistent radiation-induced electrical substrate reprogramming are important to our current study. To address this point, we have provided extensive new data elucidating the role of radiation-induced Notch signaling in conduction reprogramming after RT.

Notch signaling is known to play a role in the radiation response and radioresistance of several tissue types. In our unbiased RNA sequencing, the Notch pathway was among our most-overexpressed hallmark pathways in murine hearts at both 2w and 6w post-IR. Previous data from the Rentschler lab demonstrated that reactivation of developmental signaling pathways, including Notch, can electrically remodel adult ventricular cardiomyocytes through transcriptional and epigenetic changes that result in a conductive “Purkinje-like” phenotype (Rentschler *et al. Circ Res*, 2012).

In our revised manuscript, we provide new analysis of our RNA seq data to show upregulation of Notch signaling and differential expression of several Notch signaling pathway members (manuscript **Fig. 5**). We further provide new data to demonstrate that transient Notch activation in adult mice, mimicking a single dose of radiotherapy, persistently increases longitudinal conduction velocity and upregulates Nav1.5 for at least 16 to 52 weeks after activation (manuscript **Fig. 6**). This novel mechanistic finding has importance for our understanding of a mechanism whereby a single dose of radiation could lead to persistent changes in the electrical substrate and reduce ventricular tachycardia through cardiac reprogramming.

The following parts of the Results has been substantially revised:

“ . . . Within the most-upregulated response pathways, we also explored signaling responses previously implicated in the modulation of cardiac conduction. The Notch signaling pathway,

which plays an important role in conduction system development^{30–32}, was present at both timepoints (**Fig. 5a**). Furthermore, leading edge analysis showed that Notch pathway-associated genes were drivers of enrichment in other enriched pathways including *Cdkn1a* (8 gene sets at 6 weeks, 4 gene sets at 2 weeks), *Jag1* (4 gene sets at 2 weeks), *Jag2* (3 gene sets at 2 weeks), and *Notch2* in (3 gene sets at 2 weeks). Given that Notch signaling is known to be quiescent in resting adult cardiomyocytes, we investigated in detail the differential expression of all genes associated with the murine Notch pathway gene ontology (GO) term (**Fig. 5b**). Unsupervised hierarchical clustering of Notch GO genes ordered our samples into their respective timepoints, and each pairwise comparison of timepoints showed numerous changes in Notch GO term expression, with a trend towards Notch activation at later time points. Given this broad Notch activation signal, we also investigated gene expression in specific Notch pathway functional classes using genes with known relevance to cardiomyocytes, which demonstrated coherent Notch activation after IR (**Fig. 5c**) Previously, we reported that transient cardiomyocyte Notch reactivation resulted in persistent epigenetic and electrophysiologic changes and produced a conduction-like phenotype^{32,33}. In these studies, Notch reactivation upregulated *Scn5a*, which encodes for *Nav1.5*, and increased I_{Na} -mediated cardiomyocyte membrane excitability and peak upstroke velocity. To further determine whether cardiomyocyte Notch signaling can increase cardiac conduction, we utilized the Tet-on transgenic system to transiently and selectively activate Notch in adult cardiomyocytes (inducible Notch Intracellular Domain, *iNICD*; α MHC-rtTA; *tetO_NICD*), as previously described^{33,34}. Adult *iNICD* mice were fed doxycycline chow for 3 weeks to transiently activate Notch, followed by a minimum of 16 weeks washout for *iNICD* turnover. This doxycycline pulse followed by washout was chosen to sufficiently activate Notch while mimicking a transient dose of radiotherapy. When compared to littermate controls, longitudinal CVs of *iNICD* ventricles were significantly increased after 16-week washout (**Fig. 6a, 6b**). In a separate cohort, mice were sacrificed after a 52-week washout to test whether Notch activation leads to persistent upregulation of cardiac conduction proteins. Western blotting revealed that transient activation of Notch signaling was associated with persistent increases in ventricular *Nav1.5* (**Fig. 6c**), but not *Cx43* (**Fig. 6d**). These findings suggest that transient Notch activation may partially contribute to radiation-induced persistent upregulation of the cardiac sodium channel and are important to understanding a mechanism whereby a single dose of radiation could lead to permanent changes in the electrical substrate.”

2. The findings that radiation upregulates Cx43 expression and improves conduction is not entirely novel, as cited by the authors.

Thank you for this comment. Our study seeks to unbiasedly assess mechanisms by which ionizing radiotherapy leads to conduction reprogramming. This experimental approach resulted in partial overlap with results from previous literature that studied carbon radiation. While increases in connexins were previously reported after carbon radiation, our study is the first to report durable restoration of the cardiac sodium channel after ionizing radiation to improve conduction across species. To further expand on this novel finding, we have added new data demonstrating differential transcription of Notch pathway members after ionizing radiotherapy and Notch-mediated upregulation of *Nav1.5* in order to highlight one of several potential cell signaling mechanisms that may contribute to radiation-induced electrical substrate reprogramming.

3. The antiarrhythmic effect of Na1.5/Cx43 upregulation is not demonstrated. The radiation-induced increase in conduction velocity may make the anatomic pathlength too short to sustain reentry, but no evidence is presented to this effect. The paper would be substantially strengthened if results in an animal model with inducible VT were presented.

Thank you for this suggestion. We agree that future preclinical studies examining electrophysiologic changes in large animal models of VT have important implications for treatment planning and dose prescription to further refine the clinical cardiac radiation protocol with electrical substrate reprogramming in mind. These experiments are warranted precisely by the results presented in our current study, which seeks to delineate electrophysiologic changes after radiotherapy to improve our understanding of this new treatment modality. Within our murine model system, we provide new mechanistic insights for the role of radiation-induced Notch signaling. For these purposes, rodent models allow for the ability to use genetically-engineered systems and remain valuable for understanding these mechanisms.

Indeed, ample preclinical and clinical evidence have demonstrated that radiotherapy has an antiarrhythmic effect, and novel data in this current study from RT-treated patients demonstrates that this effect cannot be explained by radiation-induced fibrosis alone. Our study provides important evidence to suggest that, in contrast to tissue destruction that was expected after high-dose radiotherapy, we observe tissue electrophysiologic *reprogramming*.

The development of large animal models of arrhythmia that accurately represent human disease remain ongoing challenges in the field, and all preclinical models for studying therapies have limitations. The existence of a consistent and physiologically-relevant model of inducible VT remains controversial (Piktel and Wilson, *Front Cardiovasc Med*, 2019). Although large animal models show promise, these species are often more susceptible to ventricular fibrillation (VF) than tachycardia (Lukacs *et al. Acta Physiol Hung*, 2012). Our group is navigating these limitations as we pursue future studies in large animal models. Proper selection and implementation of large animal models for these purposes require a substantial amount of time and investment, best served as the subject of a future manuscript.

Although future preclinical studies in large animal models are expected to help characterize the anti-tachycardic effects of Nav1.5 and Cx43 after radiotherapy, the overexpression of either sodium channels or connexins within cardiomyocytes has previously demonstrated an anti-arrhythmic effect. Indeed, adenoviral-mediated upregulation of skeletal muscle sodium channel SkM1 (Lau *et al. Circulation*, 2009; Boink *et al. Cardiovas Res*, 2012) and Cx43 (Greener *et al. J Am Coll Cardiol*, 2012) to infarcted

FIGURE 5 Heat map showing probabilities of conduction block and re-

Reference Figure 1 from Campos, F. O. et al. Factors Promoting Conduction Slowing as Substrates for Block and Reentry in Infarcted Hearts. *Biophys. J.* 117, 2361–2374 (2019).

canine and porcine hearts, respectively, reduced ventricular arrhythmia inducibility by locally increasing conduction velocity to prevent electrical reentry.

Given the known limitations to animal models of ventricular arrhythmias, recent *in silico* studies have provided further mechanistic insights to understanding the patho-electrophysiology of reentrant tachycardia. Multiple groups have shown *in silico* that locally-reduced conduction, particularly due to reduced ventricular sodium channel current, within scars is critical to sustaining ventricular arrhythmias. Prakosa *et al.* (*Nat Biomed Eng*, 2018) required a 68% reduction in peak sodium current to adequately model infarct-related VT in patients. More recently, Campos *et al.* (*Biophys J*, 2019) report that the probability of concurrent unidirectional conduction block and reentry occurring within ventricular scar are nonnegligible only in the setting of severely reduced sodium current (below a scaling factor of 0.4). Conversely, a restoration of sodium channels above this threshold is sufficient to reduce the probability of concurrent unidirectional block and reentry to near zero (**Reference Figure 1** from Campos *et al.* provided for illustration).

4. Radiation did not result in substantial fibrosis in the animal model. Previous animal studies found 25 Gy to result in at least moderate fibrosis (ref 25, for example). What accounts for this difference?

Thank you for this important observation. There are several potential factors that may account for these differences. In these studies, including (previous) reference 25, all known histologic evidence of mild to moderate fibrosis at dosages in the range of 25 Gy are reported at or after 6 months post-irradiation (Sharma *et al.* *Heart Rhythm*, 2010; Rapp *et al.* *Sci Rep*, 2019), well beyond the period when anti-tachycardic effects of radiotherapy occur clinically but within the timeframe when radiation-induced fibrosis was expected. We specifically examined fibrosis at the acute timepoint of 6 weeks post-IR to determine whether radiation-induced fibrosis could occur during this clinically-relevant timepoint.

Additionally, many of these preclinical studies, including reference 25, examined fibrosis in supraventricular structures (pulmonary trunk, left atrial appendage, atrioventricular node) after radiation, while this studied tested changes in fibrosis within the ventricles post-IR. Clinically, RT has shown clinical efficacy only for treatment of ventricular tachycardia, and it is currently unknown whether there are different chamber-specific responses to high-dose radiation.

5. The narrowing of QRS is presented for a single patient. The authors presumably have access to a much larger sample size. Presenting what happens to the QRS for the ENCORE-VT cohort would be more convincing.

Thank you for this input. It is not our intention to suggest that all RT patients would be expected to exhibit QRS shortening. Rather, we provide here a translational example of a phenomenon observed in our murine studies. Importantly, the QRS interval is a surrogate for conduction velocity and represents total left- and right- ventricular activation time. Each patient is prescribed a target volume and location specific to the individual's scar-related VT, and localized increases in conduction are not always expected to translate into shortened QRS intervals. We highlight

QRS shortening in one patient, while acknowledging that this phenomenon would not be expected in all patients. Although not all patients in the ENCORE-VT cohort exhibited ECG changes, the shortened QRS presented in this manuscript is 1 of 4 instances of QRS shortening in this 19-patient cohort. In this study, we do not report ECG changes after RT for the entire ENCORE-VT cohort, as these effects are diverse and the subject of a different manuscript. Indeed, future clinical and preclinical studies are expected to be better powered to discern differences and define relationships among target locations, volumes, and ECG changes. We have now added this as a limitation of the current manuscript and a future direction in the Discussion.

6. The change in QRS duration observed in the animal models is relatively small, corresponding to relatively small changes in conduction that seem both inconsistent with the substantial changes in both Nav1.5 and Cx expression.

Thank you for this comment. At baseline, cardiac conduction proteins are saturated, and increases to either the sodium current (by Nav1.5) or intercellular coupling (by Cx43) no longer result in linear increases in cardiac conduction (Shaw and Rudy. *Circ Res*, 1997; Kucera, Rohr, and Rudy. *Circ Res*, 2002) (Reference Figure 2, notice the logarithmic behavior on a linear X-axis approaching 100%; Reference Figure 3, notice the linear behavior on a logarithmic X-axis approaching 100%). Although the measured magnitudes of change in QRS and CV appear relatively small, these are consistent with known logarithmic relationships between CV and sodium current, and between CV and intercellular coupling, at baseline levels of conduction protein expression. Thus, as sodium current and/or connexins approach or exceed physiologic levels, significantly greater increases in functional protein expression are required to achieve small biologic increases in electrical conduction.

Reference Figure 2 Conduction velocity (solid line) and safety factor of conduction (dashed line) versus % \bar{g}_{Na} .

Figure referenced from Shaw R.M. and Rudy Y. Ionic mechanisms of propagation in cardiac tissue. Roles of the sodium and L-type calcium currents during reduced excitability and decreased gap junction coupling. *Circ Res*. 1997 Nov;81(5):727-41.

Reference Figure 3 Conduction velocity as a function of gap junction coupling.

Figure referenced from Kucera J.P., Rohr S., Rudy Y. Localization of sodium channels in intercalated disks modulates cardiac conduction. *Circ Res*. 2002 Dec 13;91(12):1176-82.

7. What is the difference between the data in Fig 3c and Fig 3n of the paper?

Figures 3a through 3k (including 3c) are performed at the clinically-relevant timepoint of 6 weeks post-IR. Figures 3l through 3o (including 3n) are performed at 42 weeks post-IR to highlight the durability of effect. We have modified the Figure format (now Figure 2) to make this demarcation clearer to the reader.

The authors show differences between QRS duration of control and irradiated mouse hearts. What were the QRSs before irradiation? Were they comparable at that time point in the 2 groups?

Thank you for raising this important concern. We agree that it is important to show that the QRS intervals among littermates were comparable at baseline. We have provided the baseline ECG intervals for the dose de-escalation cohort in Extended Data Figure 6 to show that all pre-IR ECG intervals, including the QRS, were comparable between experimental groups.

8. Data should be shown to support the functional significance of the Nav1.5 upregulation in terms of Na current in irradiated vs control mouse cardiomyocytes.

Thank you for this comment. In this study, we demonstrate the novel finding that radiation-mediated increases in both Nav1.5 and Cx43 are sufficient to significantly increase cardiac conduction as a plausible mechanism for preventing electrical reentry after radiotherapy. In the revised manuscript, we show that Notch-mediated upregulation of Nav1.5 is one of several potential mechanisms that may contribute to electrical reprogramming after irradiation. We have previously published that Notch-mediated upregulation of the cardiac sodium channel in left ventricular cardiomyocytes significantly increased I_{Na} -mediated maximum voltage upstroke by approximately 25% through whole-cell current clamp recordings (Khandekar *et al. Circ Res*, 2016).

9. The authors show evidence of accelerated conduction in the normal myocardium outside the scar zone, which they attribute to increased Na current and Cx function. What is the evidence that these effects can restore conduction in the peri-infarct regions of conduction block that lead to scar-related VT?

Thank you for this question. We implemented a large infarct model to show that radiation-induced changes in Nav1.5 and Cx43 occur specifically in surviving cardiomyocytes and not scar myofibroblasts. In the setting of scar-related VT, the peri-infarct regions that lead to VT are comprised of channels of surviving cardiomyocytes interspersed though nonconducting scar myofibroblasts, leading to heterogeneous, slow conduction. Previous studies demonstrated that restoration of conduction proteins in these conduits of surviving cardiomyocytes is sufficient to increase conduction velocity and prevent VT. For example, adenoviral delivery of the skeletal muscle sodium channel SkM1 to border zone myocardium of infarcted canine hearts (Lau *et al, Circulation*, 2009; Boink *et al. Cardiovas Res*, 2012) and delivery of Cx43 to border zone myocardium of infarcted porcine hearts (Greener *et al. J Am Coll Cardiol*, 2012) both resulted in improved conduction and decreased inducibility of ventricular arrhythmias.

In silico, Campos *et al.* (*Biophys J*, 2019) report that the probability of concurrent unidirectional conduction block and sustained reentry within these peri-infarct regions is near-zero once sodium current is restored above approximate 40-50%, and, above this threshold, fibrosis contributes minimally to affect the probability of reentry (**Reference Figure 1**). Data from our murine infarction model suggest that, within surviving cardiomyocytes in the infarcted ventricles, protein levels of Nav1.5 more than double at 6 weeks post-IR.

Reviewer #2 (Remarks to the Author):

In this report, Zhang and colleagues examine the effects of cardiac radioablation on electrophysiologic properties using various methods. The observation that has driven this study is an observed incongruity between predicted pro-fibrotic effects and the clinically observed antiarrhythmic effects of cardiac radioablation. The authors examine post-mortem/transplant specimens from patients who have undergone cardiac radioablation, and they studied a murine model of cardiac radioablation.

The key findings are: lack of significant fibrosis seen in the irradiated myocardium, and distinct changes in conduction properties, manifest as increased conduction velocity, with observed upregulation of key cellular conduction protein components, including Nav1.5 and CX43. Additional dose finding studies suggest the possibility that lower doses of radiation may provide similar electrophysiologic tissue effects.

These findings provide a plausible mechanism for clinical observations of relatively early antiarrhythmic effects. The authors are to be applauded for performing a comprehensive, multi-modality approach to studying this phenomenon.

There remain a few questions in this reviewer's mind that would benefit from a response:

Thank you for your thoughtful assessment of our manuscript. We have provided pointwise responses to questions, below.

• The challenge of the post-mortem/transplant portion of the study is the limited number of samples, which is understandable. In addition however, the internal "control" of non-irradiated remote portions of ventricular myocardium very likely received some radiation exposure. It would be important to specify, based on isodose curves, the estimated dose received by these "control" samples. If in fact these tissues received anything above 15Gy, as the authors themselves have noted, it is likely these areas should have been expected to exhibit similar electrophysiologic changes.

Thank you for raising this important concern. We agree. At the time of specimen collection, each patient's heart is compared to its treatment plan, and an internal control is selected such that it previously received little to no radiation exposure. Across all samples, internal controls received less than 5 Gy. To address this point, we have added this clarification to the Methods and Results

sections: “Across all samples, selected remote regions of myocardium received radiation exposure that was less than 5 Gy.”

As an example, we have provided Patient F’s radiation treatment plan in Extended Data Figure 10 to show that the selected control received little to no exposure relative to the targeted region.

• **How can localized effect on target result in global ECG changes? It is entirely understandable how in the whole-heart irradiated murine model global ECG changes are seen, as the entire heart is exposed to proposed threshold radiation dosing. However, Sub-25Gy dosing, and especially sub-15Gy dosing is presumably given to the rest of ventricle? How then can localized cellular EP changes result in global ECG changes as suggested by the authors in their sample patient’s ECG? It would seem entirely plausible that this was an unrelated observation, unless there can be demonstration of global cellular changes to most of the ventricle due to a large mass of supra-threshold exposed myocardium.**

Thank you for these thoughtful questions. These questions raise an important point that, because different patients are treated with different volumes and target areas, not all patients would be expected to exhibit QRS shortening. QRS interval is a surrogate for conduction velocity and represents total left- and right- ventricular activation time. Indeed, localized increases in conduction are not always expected to translate into a shortened QRS for everyone, and it is not our intention to suggest that all RT patients would be expected to exhibit QRS shortening. We have added this statement in the discussion. In this study, we provide a single translational example of a phenomenon observed in our murine studies. We expect that future clinical and preclinical studies will define relationships among target location, target volumes, isodose curves, and ECG changes.

• **To further expand on the above two points, in a larger heart model, geographic sampling to correlate isodose delivery to cellular observations would be both enlightening and**

potentially clinically relevant in terms of expectations for treatment outcomes and cardiac effects.

Thank you for this suggestion. We agree that, based off the results of this current study, future experiments to understand whether different isodoses lead to heterogeneous cellular and electrophysiologic effects are expected to have important clinical implications. Unfortunately, our current study was not designed to further investigate this question, but we provide n=1 from an ENCORE-VT patient in this manuscript showing differences in Nav1.5 between regions. These questions are worthy of an entirely new study to elucidate potential differences among isodoses. We have added these as potential future directions in the Discussion:

“In our preclinical study, we profiled the radiation response after whole-heart radiation, while clinically, cardiac RT is limited to targeting regions of arrhythmogenic reentry. Indeed, future preclinical and clinical studies are expected to elucidate relationships among treatment target volume, isodoses, local changes in conduction protein expression, and global changes in ECG intervals.”

• It would be very reasonable, and perhaps enlightening to spend a sentence or two discussing hypotheses in more detail for why increased CV may result in antiarrhythmic effects. Possible mechanisms may include reduction of conduction Safety Factor (see Shaw and Rudy – authors’ institution!, 1985) from creation of source-sink mismatch, possibly through recruitment of surrounding myocardium. Alternatively, loss of a function zone of slow conduction may functionally eliminate the critical element needed to sustain reentry. Further discussion on this is certainly at least hypothesis generating.

Thank you for this valuable suggestion. We agree. We have further expanded the dialogue on why increases in connexins and sodium channel are expected to prevent arrhythmias in the Discussion. Although our data suggests that the mechanism is likely due to functionally eliminating electrical reentry and reduction in the vulnerable window for unidirectional block, we also discuss the potential of source-sink mismatch and reduced Safety Factor for completeness.

Local increases in sodium channels and connexins are known to result in local increases in conduction velocity, thereby eliminating regions of slow conduction that are necessary for reentry. Furthermore, reduced membrane excitability and intercellular coupling both contribute significantly to vulnerability of unidirectional block (Shaw and Rudy. *J Cardiovasc Electr*, 1995) prior to reentry, and increases to sodium current and connexins would be expected to ameliorate conditions necessary for both conduction block and reentry.

In theory, increased connexin-mediated electrical coupling may also increase the functional sink, create a source-sink mismatch, and reduce the Safety Factor to terminate arrhythmic conduction. However, this mechanism is less probable given that (1) in the setting of constant source current, mismatch from a 3-fold increase in intercellular coupling is not expected to sufficiently decrease the Safety Factor to terminate conduction (Reference Figure 4, from Shaw and Rudy. *Circ Res*, 1997), and (2) in radiation-induced electrical reprogramming, we observe both increases to the intercellular coupling “sink,” and increases in the sodium current “source”. Parallel increases to

both the source and sink are expected to further decrease the likelihood of creating adequate a source-sink mismatch alone.

Reference Figure 4 “Conduction parameters for decreased intercellular coupling. A, Conduction velocity (solid line) and safety factor for conduction (dashed line) versus gap junction conductance, g_j .”

Figure referenced from Shaw R.M. and Rudy Y. Ionic mechanisms of propagation in cardiac tissue. Roles of the sodium and L-type calcium currents during reduced excitability and decreased gap junction coupling. *Circ Res.* 1997 Nov;81(5):727-41.

The relevant text from the Discussion section is copied below:

“The functional effects of increased $Na_v1.5$ and $Cx43$, resembling electrical substrate reprogramming, may act through several electrical mechanisms to prevent conduction block and reentry and explain the antiarrhythmic effect conferred to patients after 25 Gy RT. In the setting of heart disease and conduction heterogeneity, homogenization of conduction through restoration of either membrane excitability or intercellular coupling would reduce the vulnerable window that allows unidirectional block to occur⁵⁴. Additionally, increased intercellular coupling alone may also prevent aberrant conduction from occurring by increasing the functional sink and reducing the Safety Factor to terminate arrhythmic conduction⁵¹. However, this phenomenon requires $Cx43$ increases that greatly exceed the magnitude observed after RT in our study. Finally, data from our current study suggests that, within surviving cardiomyocytes in the peri-infarct region, upregulation of $Na_v1.5$ and $Cx43$ result in local increases in conduction velocity to functionally rescue regions of slow conduction necessary for sustained reentry and tachycardia⁵¹. Future studies are warranted to determine which molecular effects can be best-leveraged in the setting of RT to prevent reentry and tachycardia.”

• **The dose finding studies are an interesting result – one concern is of course if loss of beneficial antiarrhythmic effect is observed. Would testing of varying doses then be warranted clinically?**

Thank you for highlighting this important observation. We completely agree that these results warrant future clinical studies of careful downward dose titration to find a minimum effective dose in patients. This insight is highlighted in the Discussion. Interestingly, there was an unpublished report of reduced VT burden after radiation treatment with only 20 Gy radiation in a patient in Europe (data previously presented at the 2019 Symposium for Noninvasive Radioablation in Saint Louis, MO). Our institution has recently received an Investigational Device Exemption for a multi-institutional, dose de-escalation study in the ENCORE-VT population.

• **There is also a clinical observation in multiple centers’ outcomes of late VT recurrence (6-12month range) in some patients. How would the authors hypothesize an explanation for these observations given the findings of this study?**

Thank you for pointing out these emerging clinical data. Although most patients continue to experience reduced VT burden and improved quality of life for at least 2 years after radiotherapy treatment, there are reports of significant VT recurrence in some patients that occurs months after treatment. One cause for this recurrence in a subset of individuals could be attributed to progression of cardiac disease, leading to new VTs.

Previously, new VTs were reported in 79% of patients who experienced VT recurrence after catheter ablation (Yokokawa *et al. J Am Coll Cardiol.* 2013). Indeed, many cardiac RT patients have previously failed 1 or more catheter ablations. It is currently unknown whether VT recurrence after RT originates from previously-irradiated myocardium or from new and/or previously unidentified VT exit sites. Additionally, it is unclear specifically which factors and clinical characteristics cause some patients to experience VT recurrence while others do not. Future studies, including randomized clinical trials, are expected to clarify these discrepancies and better identify which patients are best-suited for cardiac RT.

Minor point:

• **Line 58 “especially those with ??? ventricular function and ischemic cardiomyopathy with focal scar” – there is a missing word here - ?relatively preserved?**

Thank you for pointing out this error. This sentence was meant to say “. . . *especially those with preserved ventricular function . . .*”. We have made this correction in the main text.

Reviewer #3 (Remarks to the Author):

This is a very thorough and well-written article with nice illustrations. I have few comments. Please address those pointwise.

Thank you for the positive review of our study. Please see our pointwise responses below.

Earlier studies have already shown that much higher dose of radiation (40-160 Gy) is needed to induce transmural fibrosis. Therefore, the first part of the study design (to show if 25 Gy causes transmural lesions) has no novelty. Please discuss.

Thank you for emphasizing the important concept that much higher doses of radiation than those used clinically are necessary to replicate complete myocardial ablation. There is (1) a human component and (2) a murine component to the first part of the study design in question. Please see our responses regarding each component below.

(1) Human component: Currently, radiation-induced fibrosis remains a competing hypothesis for the antiarrhythmic mechanism of cardiac RT. Important preclinical evidence demonstrated that doses in excess of 40 Gy cause robust fibrosis that interrupts conduction (Lehmann *et al. Circ Arrhythmia Electrophysiol*, 2015; Suzuki *et al. Circ Arrhythmia Electrophysiol*, 2020), but it remains disputed whether the 25 Gy dose, used clinically, can produce any fibrotic lesions. As an internal example, Reviewer #1 commented that other preclinical models have demonstrated some moderate fibrosis in the 25 Gy dose range. This ambiguity likely arises, in part, from the

sigmoidal response of radiation-induced fibrosis, with doses in the range of 25 Gy near the inflection point of the dose response curve (Sharma *et al. Heart Rhythm*, 2010). The 25 Gy dose used clinically to-date was prescribed based on some preclinical observations that doses in the range of 25-35 Gy, at minimum, were required produce fibrosis on supraventricular structures.

Emerging preclinical and clinical reports continue to cite tissue destruction and fibrosis as the preferred biologic effect after cardiac radiation. These studies often discuss whether a prescription of 25 Gy results in underdosing to the target tissue, despite clinical evidence of efficacy at these lower doses. Additionally, all studies that have thoroughly evaluated radiation-induced myocardial fibrosis to-date have occurred exclusively in preclinical models. In agreement with previous preclinical studies, the data presented in Figure 1 represent the first cohort from human specimens to demonstrate that 25 Gy radiation is insufficient to cause catheter ablation-like scar, despite substantial reductions in VT burden in all patients. Figure 1 presents potentially paradigm-shifting evidence to suggest that fibrosis alone cannot account for the magnitude of VT reduction in patients treated to-date, and dose strategies may not necessarily be chosen based on the anticipated effects of late-stage, radiation-induced fibrosis. For these reasons, we believe the results in Figure 1 have direct clinical relevance and have kept these as a main result in the revised manuscript.

(2) Murine component: We agree that reproduction of this result in a murine preclinical model has no novelty. However, because fibrosis can influence intercellular coupling and conduction, these murine fibrosis results are important to support the electrophysiologic findings in our study. Furthermore, the observation of parallel activation of both pro- and anti-fibrotic response pathways at 25 Gy are important for the understanding of why fibrosis is not present at these lower doses. For these reasons, we have moved our murine fibrosis stress pathway analysis, trichrome, and hydroxyproline data from the main text to Extended Data Figures 3 and 9. Alternatively, we added new transcriptomic analyses in main text Figure 5 to elucidate the role of radiation-induced cardiac Notch signaling.

The text from the Results related to changes are copied below:

*“Consistent with preclinical studies in other animal models, we did not detect significant amounts of blue/green collagen staining in either treatment condition after 6 weeks (**Extended Data Fig. 3a**). Furthermore, we observed no difference in collagen content between treated and untreated murine ventricles 6 weeks post-IR, as determined by the more sensitive and quantitative hydroxyproline assay (**Extended Data Fig. 3b**). . . . In Gene Set Enrichment Analysis using MSigDB hallmark gene sets, canonical stress response pathways were elevated at both time points (**Fig. 5a**). We also observed that many of the most differentially-expressed genes were previously implicated in stress responses to radiation (**Extended Data Table 2**), and many of these radiation-associated genes have known pro- or anti-fibrotic functions in the heart (**Extended Data Figure 9c, 9d**). . . . Within upregulated response pathways, we also explored signaling responses previously implicated in the modulation of cardiac conduction. The Notch signaling pathway, which plays an important role in conduction system development³⁰⁻³², was significantly upregulated at both timepoints (**Fig. 5a**).”*

Sample size is very small. Thus, it is difficult to generalize the findings, which is a major limitation of the paper.

Thank you for this comment. Cardiac RT is a new antiarrhythmic therapy designated for patients with high-risk, refractory VT who have previously failed standard AAD therapy and CA or have contraindications to CA. The total number of patients who have received this therapy remain limited to-date. Fortunately, most patients survive with reduced VT burden for years after treatment, and post-mortem patient samples are even more limited. We agree these patient samples are limited, and we have added a statement addressing this limitation to our Discussion.

This limitation in available patient specimens further emphasizes the importance of the preclinical work presented in this study. For our murine studies, sample sizes were chosen to appropriately detect electrophysiologic effect sizes of 0.25 at 5% significance and at 80% or greater power. Important preclinical publications to-date have shown that very high doses in the range of 40-160 Gy are necessary to create “ablative” lesions to the heart. We believe that our study contributes significantly to these existing preclinical studies and remains one of the largest preclinical studies in this new field. Specifically, our findings expand considerably the body of evidence for radiation induced electrical substrate reprogramming, at clinical doses in the range of 25 Gy, to improve our understanding of this emerging antiarrhythmic therapy.

Was MRI used in all to define myocardial fibrosis in all?

Thank you for this question. Ten out of 19 patients in the ENCORE-VT clinical trial received contrast-enhanced MRI to define myocardial fibrosis (Robinson *et al. Circulation*, 2019). The remaining 9 patients had specific contraindications to MRI (for example, implanted ICD).

Was Bonferroni correction performed to calculate the significance for the 509 genes?

Thank you for this question. An adjustment for multiple hypothesis testing is always necessary for high-dimensional data such as RNA-seq. We limited false discovery rather than family-wise error given small sample groups available for this study. Specifically, we used Benjamini and Hochberg false discovery rate set at 5% and reported significance as adjusted *P*-values. We have clarified these further in the Methods and Results.

Please provide the baseline characteristics of the 4 patients including BMI and comorbidities.

We have added the baseline characteristics of BMI and comorbidities from all patients to Extended Data Table 1.

Depending on their observation that conduction velocity increased with shorter QRS complex in hearts treated with 15 Gy, are the authors going to change the standard dose from 25 to 15 Gy in their clinical practice?

Thank you for raising this important question. We believe these findings have important clinical implications. Prior to changing the standard dose, these results warrant future clinical studies of

careful dose titration to find a minimum effective dose in patients. We have highlighted these insights in the discussion.

References Cited in Response to Reviewers

1. Rentschler, S. *et al.* Myocardial notch signaling reprograms cardiomyocytes to a conduction-like phenotype. *Circulation* **126**, 1058–1066 (2012).
2. Lau, D. H. *et al.* Epicardial border zone overexpression of skeletal muscle sodium channel SkM1 normalizes activation, preserves conduction, and suppresses ventricular arrhythmia an in silico, in vivo, in vitro study. *Circulation* **119**, 19–27 (2009).
3. Boink, G. J. J. *et al.* SkM1 and Cx32 improve conduction in canine myocardial infarcts yet only SkM1 is antiarrhythmic. *Cardiovasc. Res.* **94**, 450–459 (2012).
4. Greener, I. D. *et al.* Connexin43 gene transfer reduces ventricular tachycardia susceptibility after myocardial infarction. *J. Am. Coll. Cardiol.* **60**, 1103–1110 (2012).
5. Prakosa, A. *et al.* Personalized virtual-heart technology for guiding the ablation of infarct-related ventricular tachycardia. *Nat. Biomed. Eng.* **2**, 732–740 (2018).
6. Campos, F. O. *et al.* Factors Promoting Conduction Slowing as Substrates for Block and Reentry in Infarcted Hearts. *Biophys. J.* **117**, 2361–2374 (2019).
7. Piktel, J. S. & Wilson, L. D. Translational Models of Arrhythmia Mechanisms and Susceptibility: Success and Challenges of Modeling Human Disease. *Frontiers in Cardiovascular Medicine* **6**, 135 (2019).
8. Lukács, E. *et al.* Overview of large animal myocardial infarction models (review). *Acta Physiologica Hungarica* **99**, 365–381 (2012).
9. Sharma, A. *et al.* Noninvasive stereotactic radiosurgery (CyberHeart) for creation of ablation lesions in the atrium. *Hear. Rhythm* **7**, 802–810 (2010).
10. Rapp, F. *et al.* Biological Cardiac Tissue Effects of High-Energy Heavy Ions – Investigation for Myocardial Ablation. *Sci. Rep.* **9**, (2019).
11. Shaw, R. M. & Rudy, Y. Ionic mechanisms of propagation in cardiac tissue: Roles of the sodium and L-type calcium currents during reduced excitability and decreased gap junction coupling. *Circ. Res.* **81**, 727–741 (1997).
12. Kucera, J. P., Rohr, S. & Rudy, Y. Localization of sodium channels in intercalated disks modulates cardiac conduction. *Circ. Res.* **91**, 1176–1182 (2002).
13. Khandekar, A. *et al.* Notch-Mediated Epigenetic Regulation of Voltage-Gated Potassium Currents. *Circ. Res.* **119**, 1324–1338 (2016).
14. Shaw, R. M. & Rudy, Y. The Vulnerable Window for Unidirectional Block in Cardiac Tissue: Characterization and Dependence on Membrane Excitability and Intercellular Coupling. *J. Cardiovasc. Electrophysiol.* **6**, 115–131 (1995).
15. Yokokawa, M. *et al.* Reasons for recurrent ventricular tachycardia after catheter ablation of post-infarction ventricular tachycardia. *J. Am. Coll. Cardiol.* **61**, 66–73 (2013).
16. Suzuki, A. *et al.* Catheter-free Arrhythmia Ablation using Scanned Proton Beams: Electrophysiologic Outcomes, Biophysics and Characterization of Lesion Formation in a Porcine Model. *Circ. Arrhythmia Electrophysiol.* **13**, 8838 (2020).
17. Lehmann, H. I. *et al.* Atrioventricular node ablation in langendorff-perfused porcine hearts using carbon ion particle therapy. *Circ. Arrhythmia Electrophysiol.* **8**, 429–438 (2015).
18. Robinson, C. G. *et al.* Phase I/II Trial of Electrophysiology-Guided Noninvasive Cardiac Radioablation for Ventricular Tachycardia. *Circulation* **139**, 313–321 (2019).

REVIEWER COMMENTS

Reviewer #1 (Remarks to the Author):

This paper nicely shows that in this model Nav1.5 and Cx43 expression are upregulated, and provide suggestive data that this effect may be mediated by Notch signaling. They also show a lack of fibrosis. Based on data from a single patient which they acknowledge are not typical, they claim that "Clinically, RT is associated with persistent increases in Nav1.5 protein expression and a shortened QRS interval on electrocardiogram (ECG)."

The authors have provided some response to prior concerns by showing that enhancing Notch signaling slightly increases conduction velocity and Nav1.5 protein expression in rat hearts. Cx43 density is said not to be increased with enhanced Notch signaling, but with large variability in the control measurements and n=3 in each group, the power to detect a physiologically meaningful e.g. 20% increase must be extremely low.

The authors essentially blow away the lack of antiarrhythmic demonstration by talking about large animal models of VT. Many studies have assessed ventricular tachycardias in rats. A Medline search revealed 344 papers under the search term "rat "ventricular tachycardia" model".

The author answer regarding QRS changes in the clinical cohort is also disingenuous. They now mention that 4 of 19 patients show QRS shortening and that "these effects are diverse" and yet highlight results from a single patient as a "proof of principle".

They similarly fail to address the question about changes in conduction in the rat peri-infarct border zone. Instead they show that the changes in Nav1.5 and Cx43 occur in cardiomyocytes and not fibroblasts, which is hardly a shock.

Reviewer #2 (Remarks to the Author):

This reviewer appreciates the thoughtful responses to comments. The remaining concern are the statements about effects of radiation on global conduction properties manifest as QRS narrowing. It still is the opinion of this reviewer that these observations are probably too unclear to draw any conclusions, and perhaps this section should be omitted from the manuscript.

Reviewer #3 (Remarks to the Author):

The authors have addressed my comments satisfactorily. I have no further comments

Author Response to Reviewers:

Thank you for continuing to review our manuscript and for the opportunity to resubmit following our Provisional Response to Reviewers. In this revision, we have completed key experiments that further solidify the role of Notch in mediating radiation-associated increases in conduction velocity. We have included a detailed discussion below of all available preclinical model systems for VT and included the rationale for our current choice, limitations of this choice, and plans for future study. We have increased the rigor of our Notch Western blot study by increasing the sample numbers as requested. We have included additional human ECG data as requested by the reviewers and editors. Below is a point-by-point response to the most recent review.

Reviewer #1 (Remarks to the Author):

This paper nicely shows that in this model Nav1.5 and Cx43 expression are upregulated, and provide suggestive data that this effect may be mediated by Notch signaling. They also show a lack of fibrosis. Based on data from a single patient which they acknowledge are not typical, they claim that "Clinically, RT is associated with persistent increases in Nav1.5 protein expression and a shortened QRS interval on electrocardiogram (ECG)."

We agree that this claim in the abstract is too bold given the data available. We have toned down this sentence to specify that 1 out of 1 available explanted heart exhibited radiation-associated increased Nav1.5 protein expression and specified that 4 out of 19 total patients in the ENCORE-VT study displayed RT-associated QRS shortening on ECG. We have also provided the complete ECG data for the for patients from whom a QRS shortening was observed in the manuscript and Extended Data.

The modified abstract text is copied below:

"Clinically, RT was associated with increased Nav1.5 expression in 1 of 1 available explanted heart and shortened QRS intervals on electrocardiogram (ECG) in 4 of 19 patients."

The authors have provided some response to prior concerns by showing that enhancing Notch signaling slightly increases conduction velocity and Nav1.5 protein expression in rat hearts. Cx43 density is said not to be increased with enhanced Notch signaling, but with large variability in the control measurements and n=3 in each group, the power to detect a physiologically meaningful e.g. 20% increase must be extremely low.

Thank you for this insightful comment. Since our observed effect sizes in conduction proteins from previous results were greater than 50%, our immunoblots were not powered to test for smaller differences. We have re-tested whether Cx43 changes exist with enhanced Notch signaling using additional mouse samples at an n=6 in each group, which was previously chosen for our conduction velocity experiments to detect an effect size of 25% at 80% power. We still found that there was no significant difference between groups at this power (Figure 6d, 1.05 ± 0.14 versus 1.00 ± 0.11 , $P = 0.47$, $n = 6$). This selected sample size was also sufficient to detect a 20% difference at 70% power.

The authors essentially blow away the lack of antiarrhythmic demonstration by talking about large animal models of VT. Many studies have assessed ventricular tachycardias in rats. A Medline search

revealed 344 papers under the search term "rat "ventricular tachycardia" model".

Thank you for highlighting the contributions of rat models of VT. It is important to note that all preclinical models or cardiac arrhythmia have important limitations, and we have included a discussion of these limitations, along with the rationale for the choice of our current model system, in the revised manuscript. We used mice in this study, as this model system is better suited to genetic manipulation, which can be used to test observed associations for direct causality of phenotypes. We have now taken advantage of this capability, by including the results of RT in a cardiomyocyte-specific, inducible Notch loss-of-function (Notch iLOF; *Myh6-MerCreMer*; *R26R^{dnMAML}*). We feel this result will significantly strengthen the current manuscript.

We do not believe that a rodent (mouse or rat) model of VT would be suitable for specifically demonstrating an antiarrhythmic effect in the context of this manuscript because specifically for acquired VTs, such as those treated with RT in the ENCORE-VT study, there are important differences in cellular and ionic pathophysiology that predispose to arrhythmias in large animals and rodents, including mice and rats, as delineated below. As such, additional studies in rats will not alleviate some of the limitations of our current model system, which we plan to address in the future using porcine models.

In both human and large-animal models (dog, pig, sheep, rabbit) of acquired arrhythmia, heart failure-associated decreases in either expression or function of the cardiac sodium channel¹⁻⁵ or gap junctions^{6,7} lead to conduction heterogeneity and impaired impulse propagation, which predisposes to self-sustaining reentry⁸ and infarct-related VT^{2,3,5}. Studies estimate reductions in peak sodium current ranging between 40-70% in both human⁹ and large animal models^{2,4} of VT. Indeed, to adequately test the antiarrhythmic effect of radiation-induced restoration of conduction velocity and conduction proteins highlighted in our study, a clinically-relevant animal model at minimum must adequately downregulate these proteins in the setting of heart failure-associated VT (for example, a porcine model).

Unfortunately, evidence does not suggest that rodent models exhibit the same etiology of infarct-associated arrhythmia as larger animals. Although select models of heart failure and infarct-related VT exist in rats^{10,11} (but not mice), the ionic mechanisms that contribute to this phenotype are due to changes in repolarizing currents^{10,12,13}, causing early- or delayed-afterdepolarizations and triggers. The most-cited rodent study of infarct-related VT by Qin *et al.*¹⁰, which is the only study to investigate the ionic basis of rodent VT, reported repolarization heterogeneity and action potential duration (APD) prolongation due to decreased transient-outward potassium current density as the mechanism of this rodent VT. The authors concluded that changes in potassium-mediated repolarization and APD prolongation result in triggered activity to cause VT in these models. Of note, the authors in that study reported no difference in the I_{Na} -driven phase 0 voltage upstroke, suggesting no presence of sodium channel downregulation in rodent models of heart failure, an important difference relative to humans. As expected, rodent models respond best to therapies that increase potassium currents¹⁴ or decrease calcium currents¹⁵ to correct pathologic changes in APD, and we would not expect therapies that restore sodium currents, gap junctions, and conduction to be necessarily as relevant nor as effective in this context.

Although some mouse and rat models of VT in the literature target conduction proteins to achieve pro-arrhythmia, nearly all of these implement gain- or loss-of-function mutations to affect ion channel function or regulation^{16,17}, including *SCN5A* and *GJA1*, that resemble congenital arrhythmias and channelopathies such as Brugada syndrome^{18,19}, Rett syndrome²⁰, Long QT syndrome²¹, and sudden infant death syndrome²². While the application of this therapy to these congenital arrhythmias is

potentially interesting, these transgenic models do not exhibit scar and border zones and do not accurately represent the current patient population receiving cardiac radiotherapy for refractory VT (end-stage heart failure patients who are ineligible or have previously failed catheter ablation).

Thus, although rodent models provide important insights to mechanisms of arrhythmogenesis and potential therapy, the pathophysiology of these small animal models of arrhythmia are not suitable for the specific testing of anti-tachycardic effects of electrical substrate reprogramming by radiation therapy presented in our study, and all previous therapeutic studies that have sufficiently demonstrated anti-arrhythmic effects in a similar context were reported in large animal models²³⁻²⁷. Among these studies, Amino *et al.* previously demonstrated that targeted heavy ion radiation therapy was associated with decreased ventricular arrhythmia inducibility in both rabbit and dog infarct models^{23,27}. Specifically, they reported that 15 Gy radiation to microsphere-induced scar in rabbits and dogs decreased VT/VF inducibility from 80% to 20%²³ and from 100% to 25%²⁷, respectively. Furthermore, they reported that the observed antiarrhythmic effect was associated with increased Cx43 expression to restore local conduction velocity in the peri-infarct regions. Although neither study tested whether radiation affects Na_v1.5 regulation, increases in either or both ionic contributors to cardiac conduction velocity is expected to have similar conductive and therapeutic effects. Although we are unable to test a physiologically-relevant mouse model of VT to demonstrate an antiarrhythmic effect of radiation in our current study for reasons stated above, the antiarrhythmic effects of radiation in large animal models of MI, associated with Cx43 upregulation and conduction velocity, are previously-published finding and has direct relevance to our current manuscript.

At this time, with COVID restrictions in place, we are unable to establish our own large animal radiation electrophysiology research program, but plan to do so in the near future. We are hopeful that publication of the current study will allow us to successfully apply for grant funding so that we can support the significant costs associated with large animal radiation and electrophysiology studies.

Relevant sections from the revised manuscript are copied below:

“In this study, we utilized murine model systems to allow for genetic manipulation and testing of direct causality of radiation-induced conduction reprogramming. Specifically, we demonstrate that cardiomyocyte Notch activation is both sufficient to increase cardiac conduction and necessary to achieve the full effects of radiation-induced conduction reprogramming through gain- and loss-of-function approaches Furthermore, while humans and large animals exhibit HF-associated decreases in conduction protein expression and function, rodent models of heart failure exhibit remodeling of repolarizing currents, making electrical reentry and its prevention via upregulation of conduction proteins difficult to demonstrate in mice. Future studies in large animal models of VT are warranted.”

The author answer regarding QRS changes in the clinical cohort is also disingenuous. They now mention that 4 of 19 patients show QRS shortening and that "these effects are diverse" and yet highlight results from a single patient as a "proof of principle".

Thank you for this comment. The quote stating that “these effects are diverse” was not specific to the QRS interval but referred to changes across different ECG intervals. To this extent, we misinterpreted part of this Reviewer’s earlier suggestion, which was similar to a question from one other reviewer, as a request to discuss global ECG changes rather than the QRS. We agree that changes in the QRS interval may have direct relevance to the findings presented in our study, and we meant to state that reporting

changes to other ECG parameters would be purely observational, would not appropriately contribute to these findings, and should be the subject of a different manuscript.

In in our revised manuscript, we have included the ECG results of the 4 patients that showed QRS shortening (Figure 9, Extended Data Figures 12, 13, and 14), specified that these changes occur in only 4 of 19 total patients in the ENCORE-VT trial, and included the following a statement in the Discussion: “Although we observed 4 examples of QRS shortening out of 19 total patients, this observation cannot be excluded as an unrelated phenomenon in the absence of larger cohort sizes and additional analyses.”

They similarly fail to address the question about changes in conduction in the rat peri-infarct border zone. Instead they show that the changes in Nav1.5 and Cx43 occur in cardiomyocytes and not fibroblsts, which is hardly a shock.

Thank you for this helpful comment related to the peri-infarct border zone analysis. In addressing the initial rebuttal request, “What is the evidence that these effects [increased Na current and Cx function] can restore conduction in the peri-infarct regions... ?”, we felt it was helpful to highlight that multiple large animal studies have shown increased sodium channels^{25,28} or connexins²⁶ (in cardiomyocytes and not fibroblasts) to directly improve conduction near the peri-infarct border zone. Indeed, the true border zone is defined by interdigitating fibrosis and cardiomyocytes that co-exist at the perimeter of infarct scar. As demonstrated by these studies highlighted above, upregulation of these proteins in the cardiomyocyte fraction is expected to increase conduction in the peri-infarct border zone.

In most animal models, the border zone in infarct models is identified anatomically by the small segment of tissue surrounding the edge of the infarct^{29,30}. Because the exact size of this narrow segment is somewhat subjective in small animal models, we did not define a pure border zone-specific conduction velocity. However, the transverse epicardial conduction velocities reported in our manuscript include isochrone measurements from the border zones of the infarcted hearts.

From this reviewer’s comments, we have modified our Figure 4 to demonstrate that differential border zone conduction may be represented in the transverse conduction velocity values. Specifically, we have provided activation maps shown previously in manuscript **Figure 4a** at a 10ms timescale to confirm that activation isochrones at the level of the LAD ligation and peri-infarct are larger in MI+IR mouse hearts compared to MI-only controls, giving rise to the conduction velocity increases previously reported in manuscript Figure 4b. We have also specified that the transverse conduction velocity measurements reported include activation signals from the epicardial border zone and defined this border zone as the ~2mm segment of tissue above the scar adjacent to the coronary ligation.

From Figure 4a. Top: Activation maps from MI (left) and MI+IR (right) mice (from manuscript Figure 4a) at a 10ms time scale. Black arrows point to location of LAD ligation. Black box denotes approximate border zone at the scar/myocardium interface. Regions of scar that were epicardially-stimulated but did not capture are denoted by grey. Bottom: Magnified isochrones in the border zone region used to calculate transverse conduction velocities (in manuscript Figure 4b). Scale bars = 5 mm.

Summary of Additions to Manuscript Revision Using Genetically-Engineered Mouse Models:

Although mice are suboptimal for modeling acquired arrhythmia or testing anti-tachycardic effects, their use as genetically-engineered model systems provide important mechanistic insights to the biology of radiation-induced substrate reprogramming. To further develop and probe the novel finding of sodium channel upregulation in this manuscript, we propose in a revision to include our new mechanistic data obtained from experiments that tested whether myocardial Notch signaling is necessary to achieve radiation-induced reprogramming of the electrical substrate. In our previous revision, our data suggested that (1) Notch is an overrepresented signaling pathway after 25 Gy radiation, and that (2) transient activation of Notch in adult cardiomyocytes is sufficient to durably increase Nav1.5 expression and conduction velocity. These earlier findings provided us rationale to perform experiments designed to test to what extent Notch signaling is required to achieve enhanced conduction or to upregulate Nav1.5 after 25 Gy irradiation.

Notch signal transduction occurs following cleavage of the intracellular domain of any Notch receptor (NICD) and its translocation to the nucleus, where it binds to multiple transcriptional coactivators to elicit direct and indirect Notch-mediated effects. For these effects to occur, binding of NICD with a family of mastermind-like (MAML) transcriptional coactivators is necessary for formation of a functional DNA-binding complex to activate Notch-specific transcription. To determine whether Notch signaling within cardiomyocytes is required for radiotherapy-induced changes in conduction velocity, we utilized a cardiomyocyte-specific, inducible Notch loss-of-function (Notch iLOF; *Myh6-MerCreMer; R26R^{dnMAML/+}*) mouse. Notch iLOF mice utilize a tamoxifen-inducible Cre recombinase under the cardiomyocyte-specific myosin heavy chain promoter (*Myh6-MerCreMer*) to regulate expression of a dominant-negative mutant variant of MAML (dnMAML) downstream of a loxP-flanked STOP cassette. This dnMAML mutant represses transactivation and nuclear activity of the NICD for all four Notch receptors by binding to and inhibiting the NICD-CBF1/SuH/Lag2 transcriptional complex³¹. Upon tamoxifen-induced Cre recombination, Notch iLOF mice permanently express dnMAML in cardiomyocytes to effectively and specifically knock out Notch signaling in a cell-autonomous manner³². Littermates expressing *Myh6-MerCreMer*, receiving tamoxifen, and lacking the dnMAML mutant were used as controls. We expected that, if Notch signaling is not only sufficient but also necessary for radiation-induced electrical substrate reprogramming, dnMAML would rescue radiation-induced changes in Nav1.5 and conduction velocity.

The manuscript Results text relevant to the above experiments is copied below:

“To test whether cardiomyocyte Notch signaling is not only sufficient but also necessary for reprogramming conduction after radiation, we utilized cardiomyocyte-specific, inducible Notch loss-of-function transgenic mice (Notch iLOF, α MHC-MerCreMer and R26R^{dnMAML/+}). Notch iLOF mice express a tamoxifen-inducible Cre recombinase under the cardiomyocyte-specific myosin heavy chain promoter, as well as a loxP-flanked STOP cassette upstream a dominant-negative mutant of the mastermind-like Notch transcriptional coactivator (dnMAML), which represses nuclear transactivation of the Notch intracellular domain³⁶. Upon tamoxifen-induced Cre recombination, Notch iLOF mice permanently express dnMAML in cardiomyocytes to effectively and specifically knock out Notch signaling in a cell-autonomous manner³⁴. Littermates expressing α MHC-MerCreMer and receiving tamoxifen but lacking the dnMAML mutant were used as controls.

Following tamoxifen-induced Cre recombination and washout, Notch iLOF mice and littermate controls were treated with 25 Gy radiation. In non-irradiated sham mice, there was no baseline difference in CV between treatment groups (Extended Data Fig. 10a). Although radiation was associated with increases in CV, Nav1.5, and Cx43 in both groups (Extended Data Fig. 10b, 10c, 10d), Notch iLOF attenuated the post-IR conduction increase by more than 30%, resulting in significantly decreased CVs in Notch iLOF

mice compared to littermate controls (Fig. 7a, 7b, (Extended Data Fig. 10e). Following radiation, Notch iLOF mice expressed significantly decreased Nav1.5 expression (Fig. 7c), while we detected no difference in Cx43 expression (Fig. 7d, Extended Data Fig. 10f). From this observed partial rescue, these data suggest that radiation-induced electrical reprogramming is partially regulated by Notch-induced upregulation of the cardiac sodium channel.”

From Fig. 7. Loss of cardiomyocyte Notch signaling inhibits radiation-induced conduction reprogramming. (a) Representative activation maps and (b) longitudinal conduction velocity measurements of irradiated littermate control (left, CTRL) and Notch iLOF (right, $\alpha MHC-MerCreMer; R26r^{dnMAML/+}$) murine hearts at 6 weeks post-IR. * $P = 0.027$ (c-d) Western blot and quantified densitometry of Nav1.5 and Cx43 from irradiated control versus Notch iLOF murine hearts at 6 weeks post-IR. * $P = 0.012$ by two-way unpaired t -test. All

We believe that results from these experiments, together with the existing data in the manuscript showing Notch activation regulates Nav1.5 and not Cx43, would provide important mechanistic data in a revision that further implicates Notch signaling in regulating electrical reprogramming. We believe that including results from these experiments in a manuscript revision would significantly strengthen our study. A new summary figure compiling the mechanistic findings of our manuscript revision is illustrated below (Author Response Summary Figure).

Author Response Summary Figure. Summary of main findings presented in the proposed revised manuscript.

Reviewer #2 (Remarks to the Author):

This reviewer appreciates the thoughtful responses to comments. The remaining concern are the statements about effects of radiation on global conduction properties manifest as QRS narrowing. It still is the opinion of this reviewer that these observations are probably too unclear to draw any conclusions, and perhaps this section should be omitted from the manuscript.

Thank you for the positive feedback and reception of our previous responses. We have revised sections of the manuscript text concerning conclusions made from the observed QRS narrowing in the patient cohort. We have included the ECG results of the 4 patients that showed QRS shortening (**Figure 9, Extended Data Figures 12, 13, and 14**), specified that these changes occur in only 4 of 19 total patients in the ENCORE-VT trial, and included the following a statement in the Discussion: *“Although we observed 4 examples of QRS shortening out of 19 total patients, this observation cannot be excluded as an unrelated phenomenon in the absence of larger cohort sizes and additional analyses.”*

Reviewer #3 (Remarks to the Author):

The authors have addressed my comments satisfactorily. I have no further comments.

Thank you for reviewing our manuscript and for the positive reception of our responses.

References Cited in Response to Reviewers

1. Zicha, S., Maltsev, V. A., Nattel, S., Sabbah, H. N. & Undrovinas, A. I. Post-transcriptional alterations in the expression of cardiac Na⁺ channel subunits in chronic heart failure. *J. Mol. Cell. Cardiol.* **37**, 91–100 (2004).
2. Valdivia, C. R. *et al.* Increased late sodium current in myocytes from a canine heart failure model and from failing human heart. *J. Mol. Cell. Cardiol.* **38**, 475–483 (2005).
3. Nattel, S., Maguy, A., Le Bouter, S. & Yeh, Y. H. Arrhythmogenic ion-channel remodeling in the heart: Heart failure, myocardial infarction, and atrial fibrillation. *Physiol. Rev.* **87**, 425–456 (2007).
4. Maguy, A. *et al.* Ion channel subunit expression changes in cardiac purkinje fibers: A potential role in conduction abnormalities associated with congestive heart failure. *Circ. Res.* **104**, 1113–1122 (2009).
5. Dybkova, N. *et al.* Differential regulation of sodium channels as a novel proarrhythmic mechanism in the human failing heart. *Cardiovasc. Res.* **114**, 1728–1737 (2018).
6. Dupont, E. *et al.* Altered connexin expression in human congestive heart failure. *J. Mol. Cell. Cardiol.* **33**, 359–371 (2001).
7. Severs, N. J., Bruce, A. F., Dupont, E. & Rothery, S. Remodelling of gap junctions and connexin expression in diseased myocardium. *Cardiovascular Research* **80**, 9–19 (2008).
8. Campos, F. O. *et al.* Factors Promoting Conduction Slowing as Substrates for Block and Reentry in Infarcted Hearts. *Biophys. J.* **117**, 2361–2374 (2019).
9. Prakosa, A. *et al.* Personalized virtual-heart technology for guiding the ablation of infarct-related ventricular tachycardia. *Nat. Biomed. Eng.* **2**, 732–740 (2018).
10. Qin, D. *et al.* Cellular and ionic basis of arrhythmias in postinfarction remodeled ventricular myocardium. *Circ. Res.* **79**, 461–473 (1996).
11. Chinyere, I. R. *et al.* Progression of infarct-mediated arrhythmogenesis in a rodent model of heart failure. *Am. J. Physiol. - Hear. Circ. Physiol.* **320**, H108–H116 (2021).
12. de BRITO SANTOS, P. E., BARCELLOS, L. C., MILL, J. G. & MASUDA, M. O. Ventricular Action Potential and L-Type Calcium Channel in Infarct-Induced Hypertrophy in Rats. *J. Cardiovasc. Electrophysiol.* **6**, 1004–1014 (1995).
13. Ren, C. *et al.* Nerve sprouting suppresses myocardial Ito and IK1 channels and increases severity to ventricular fibrillation in rat. *Auton. Neurosci. Basic Clin.* **144**, 22–29 (2008).
14. Lebeche, D. *et al.* In vivo cardiac gene transfer of Kv4.3 abrogates the hypertrophic response in rats after aortic stenosis. *Circulation* **110**, 3435–3443 (2004).
15. Abraham, M. R. *et al.* Spiral waves and reentry dynamics in an in vitro model of the healed infarct border zone. *Circ. Res.* **105**, 1062–1071 (2009).
16. Remme, C. A. Cardiac sodium channelopathy associated with SCN5A mutations: Electrophysiological, molecular and genetic aspects. *J. Physiol.* **591**, 4099–4116 (2013).
17. Danik, S. B. *et al.* Modulation of cardiac gap junction expression and arrhythmic susceptibility. *Circ. Res.* **95**, 1035–1041 (2004).
18. Martin, C. A., Zhang, Y., Grace, A. A. & Huang, C. L. H. Vivo studies of Scn5a^{+/−}-mice modeling brugada syndrome demonstrate both conduction and repolarization abnormalities. *J. Electrocardiol.* **43**, 433–439 (2010).
19. Valdivia, C. R. *et al.* Loss-of-function mutation of the SCN3B-encoded sodium channel 3 subunit associated with a case of idiopathic ventricular fibrillation. *Cardiovasc. Res.* **86**, 392–400 (2010).
20. McCauley, M. D. *et al.* Rett syndrome: Pathogenesis of lethal cardiac arrhythmias in Mecp2 mutant mice: Implication for therapy in Rett syndrome. *Sci. Transl. Med.* **3**, 113ra125 (2011).
21. Wang, Q. *et al.* Cardiac sodium channel mutations in patients with long QT syndrome, an inherited cardiac arrhythmia. *Hum. Mol. Genet.* **4**, 1603–1607 (1995).
22. Van Norstrand, D. W. *et al.* Connexin43 mutation causes heterogeneous gap junction loss and

- sudden infant death. *Circulation* **125**, 474–481 (2012).
23. Amino, M. *et al.* Heavy ion radiation up-regulates Cx43 and ameliorates arrhythmogenic substrates in hearts after myocardial infarction. *Cardiovasc. Res.* **72**, 412–421 (2006).
 24. Lau, D. H. *et al.* Epicardial border zone overexpression of skeletal muscle sodium channel SkM1 normalizes activation, preserves conduction, and suppresses ventricular arrhythmia an in silico, in vivo, in vitro study. *Circulation* **119**, 19–27 (2009).
 25. Boink, G. J. J. *et al.* SkM1 and Cx32 improve conduction in canine myocardial infarcts yet only SkM1 is antiarrhythmic. *Cardiovasc. Res.* **94**, 450–459 (2012).
 26. Greener, I. D. *et al.* Connexin43 gene transfer reduces ventricular tachycardia susceptibility after myocardial infarction. *J. Am. Coll. Cardiol.* **60**, 1103–1110 (2012).
 27. Amino, M. *et al.* Inducibility of Ventricular Arrhythmia 1 Year Following Treatment with Heavy Ion Irradiation in Dogs with Myocardial Infarction. *PACE - Pacing Clin. Electrophysiol.* **40**, 379–390 (2017).
 28. Coronel, R. *et al.* Cardiac expression of skeletal muscle sodium channels increases longitudinal conduction velocity in the canine 1-week myocardial infarction. *Hear. Rhythm* **7**, 1104–1110 (2010).
 29. Van Duijvenboden, K. *et al.* Conserved NPPB + Border Zone Switches from MEF2- to AP-1-Driven Gene Program. *Circulation* **140**, 864–879 (2019).
 30. Janse, M. J. *et al.* The ‘border zone’ in myocardial ischemia. An electrophysiological, metabolic, and histochemical correlation in the pig heart. *Circ. Res.* **44**, 576–588 (1979).
 31. Tu, L. L. *et al.* Notch signaling is an important regulator of type 2 immunity. *J. Exp. Med.* **202**, 1037–1042 (2005).
 32. Khandekar, A. *et al.* Notch-Mediated Epigenetic Regulation of Voltage-Gated Potassium Currents. *Circ. Res.* **119**, 1324–1338 (2016).

REVIEWER COMMENTS

Reviewer #1 (Remarks to the Author):

I am satisfied with this revision, with the exception of one point.

The authors emphasize shortened QRS duration in 4 of 19 patients. They rationalize this by saying that "not all RT patients would be expected to exhibit a shortened QRS duration". To me, this is not science. I accept that not all would be expected to show reduced QRS duration, but they need to provide mean \pm SEM values for QRS duration in all patients before and after radiotherapy. This should be provided in the main paper text, along with the associated P value. They can then refer to the Extended Data figures to make their argument about the 4 showing reduced QRS duration, but they should also include in the extended data a scatter plot with all patient QRSs before and after radiotherapy, with pre- and post- values for individual subjects connected with a line so that the reader can judge the data for themselves.

AUTHOR RESPONSE TO REVIEWER COMMENTS

Reviewer #1 (Remarks to the Author):

I am satisfied with this revision, with the exception of one point.

The authors emphasize shortened QRS duration in 4 of 19 patients. They rationalize this by saying that "not all RT patients would be expected to exhibit a shortened QRS duration". To me, this is not science. I accept that not all would be expected to show reduced QRS duration, but they need to provide mean \pm -SEM values for QRS duration in all patients before and after radiotherapy. This should be provided in the main paper text, along with the associated P value. They can then refer to the Extended Data figures to make their argument about the 4 showing reduced QRS duration, but they should also include in the extended data a scatter plot with all patient QRSs before and after radiotherapy, with pre- and post- values for individual subjects connected with a line so that the reader can judge the data for themselves.

Thank you for continuing to review our manuscript and for this important feedback. We agree with the Reviewer's request and have provided the pre- and post-radiotherapy QRS durations of all 19 patients in a paired scatter plot in **Figure 8d** of the manuscript. In the main text, we have also provided the corresponding means and statistics within this cohort. Within the 19-patient ENCORE-VT cohort, we observed a small but statistically nonsignificant difference towards QRS shortening after radiotherapy (149 ± 34 ms pre-RT versus 139 ± 32 ms post-RT, $n = 19$, $P = 0.072$). Because many patients exhibited small changes in QRS duration, we have also defined, for clarification, robust QRS shortening in the 4 patients, whose ECGs were highlighted and provided in the Supplementary data, as a decrease in QRS duration by at least 25 ms on serial surface ECGs.

Revised manuscript text relevant to this response are copied below:

"Next, we analyzed serial surface ECGs. In contrast to the murine heart, where the whole heart is irradiated, only the arrhythmogenic region of a patient's heart is targeted. Thus, not all RT patients would be expected to exhibit a shortened QRS duration. Even still, we observed a statistically nonsignificant difference in QRS duration within the 19-patient cohort by 6 weeks post-RT (149 ± 34 ms pre-RT versus 139 ± 32 ms post-RT, $n = 19$, $P = 0.072$, Fig. 8d). Within this patient cohort, we report robust RT-associated QRS shortening, defined strictly as a decrease in the QRS duration by 25 ms or greater on serial ECGs, in at

Figure 8d. Matched QRS intervals in the ENCORE-VT patient cohort at baseline (pre-RT, 149 ± 34 ms) versus 6-week post-RT (139 ± 32) timepoints. $n = 19$, $P = 0.072$ by Wilcoxon signed-rank test.

least 4 out of the 19 patients (Fig. 8e, Supplementary Fig. 12-14). As an example, Patient G's QRS shortened from 165 ms with left bundle branch block (LBBB) to a QRS interval of 130 ms without LBBB, and this QRS remained shortened for at least 6 months (Fig 8e). Presented in the Supplementary Figures are the pre- and post-RT ECGs of the remaining 3 patients who exhibited persistent QRS shortening (Supplementary Fig. 12-14), as well as an example where there was no shortening observed (Supplementary Fig. 15). Consistent with our preclinical data in mice, these ECG changes may be potential examples of radiation-induced functional electrical reprogramming in humans."

REVIEWER COMMENTS

Reviewer #1 (Remarks to the Author):

I am satisfied with the addition of the figure showing QRS changes in all patients.

However, the text has not been adjusted to represent the data fairly and objectively. For example:

Abstract:

"Clinically, RT was associated with increased NaV1.5 expression in 1 of 1 explanted heart and shortened QRS intervals on electrocardiogram (ECG) in at least 4 of 19 patients."

This should be reworded to reflect the fact that the QRS changes were statistically nonsignificant, with a tendency to decrease.

While 4 patients did show a "robust" decrease, 3 showed substantial increases, and I don't feel that it is correct to emphasize the 4 that did show large QRS decreases in the abstract because it gives a biased perspective on the results.

Results:

"Within this patient cohort, we report robust...(Fig. 8E)." After this, the authors should provide some reflection of the overall patient changes like: "Overall, 9 patients showed decreases in QRS duration on follow-up, while 5 showed increases."

AUTHOR RESPONSE TO REVIEWER COMMENTS

REVIEWER COMMENTS

Reviewer #1 (Remarks to the Author):

I am satisfied with the addition of the figure showing QRS changes in all patients. However, the text has not been adjusted to represent the data fairly and objectively.

Thank you for continuing to review our manuscript and for this important feedback. We have made changes to the manuscript text to represent the data fairly and objectively. Please see our specific edits below.

For example:

Abstract:

"Clinically, RT was associated with increased NaV1.5 expression in 1 of 1 explanted heart and shortened QRS intervals on electrocardiogram (ECG) in at least 4 of 19 patients."

This should be reworded to reflect the fact that the QRS changes were statistically nonsignificant, with a tendency to decrease.

While 4 patients did show a "robust" decrease, 3 showed substantial increases, and I don't feel that it is correct to emphasize the 4 that did show large QRS decreases in the abstract because it gives a biased perspective on the results.

We agree that fair and objective representation of these data in the text is vital, and that emphasis of 4 robust patients in the Abstract may bias reader perception of the Results. We believe that the most transparent and objective representation of these data in the Abstract is a statement of how many patients experienced any shortening or lengthening of QRS duration, regardless of magnitude, similar to this Reviewer's suggested revision for the Results section. We have modified the Abstract to objectively reflect these results.

The relevant modified text from the Abstract is copied below:

"Clinically, RT was associated with increased Na_v1.5 expression in 1 of 1 explanted heart. On electrocardiogram (ECG), post-RT QRS durations were shortened in 13 of 19 patients and lengthened in 5 patients."

Results:

"Within this patient cohort, we report robust...(Fig. 8E)." After this, the authors should provide some reflection of the overall patient changes like: "Overall, 9 patients showed decreases in QRS duration on follow-up, while 5 showed increases."

Thank you for this additional feedback. We have modified the Results section to add a statement to reflect overall changes in the patient cohort on an individual basis, in which 13 patients had shorter post-RT QRS durations compared to baseline and 5 patients exhibited longer post-RT QRS durations (**Response to Reviewers Figure 1**, and Statistical Source Data

supplement file). We have included this statement before the sentence that presents examples of robust QRS shortening in the 4 patients, as it directly relates to the preceding sentence that reports nonsignificant results in the mean QRS duration of the entire cohort.

The relevant modified text from the Results section is copied below:

*“We observed a statistically nonsignificant difference in QRS durations within the 19-patient cohort by 6 weeks post-RT (149 ± 34 ms pre-RT versus 139 ± 32 ms post-RT, $n = 19$, $P = 0.072$, **Fig. 8d**). Overall, compared to baseline intervals, post-RT QRS durations were shorter in 13 patients and longer in 5 patients (**Fig. 8d**). We also present examples of robust RT-associated QRS shortening, defined strictly as a decrease in the QRS duration by 25 ms or greater on serial ECGs, in at least 4 out of the 19 patients (**Fig. 8e, Supplementary Fig. 12-14**).”*

Additional edits to the Manuscript’s Introduction and Discussion sections relevant to the representation of these results have been copied below:

Introduction:

“We also report a statistically nonsignificant difference in QRS intervals, with a tendency towards QRS shortening, in 19 patients treated with RT.”

Discussion:

“Given that each patient is prescribed a target volume and location specific to the individual’s scar-related VT, we also acknowledge that global electrophysiologic changes reflected by changes to the QRS interval is neither observed nor expected in every patient. Although we observed 13 patients with a shortened post-RT QRS interval, these differences were statistically nonsignificant in our 19-patient cohort and at least 5 patients experienced a lengthening in their post-RT QRS duration. Overall, these changes in QRS duration cannot be excluded as an unrelated phenomenon in the absence of additional analyses...”

Response to Reviewers, Figure 1.

(a) From Figure 8d in the main manuscript text: Matched QRS intervals in the ENCORE-VT patient cohort at baseline (pre-RT, 149 ± 34 ms) versus 6-week post-RT (139 ± 32) timepoints. $n = 19$, $P = 0.072$ by Wilcoxon signed-rank test.

(b) From Statistical Source Data supplementary document: Pre-RT and Post-RT values plotted in Figure 8d. Patient rows with any difference in post-RT QRS duration have been highlighted. Blue highlight represents patients with shortened QRS intervals (thirteen; $\text{Post} - \text{Pre} < 0$); orange represents lengthened QRS intervals (five; $\text{Post} - \text{Pre} > 0$).

REVIEWERS' COMMENTS

Reviewer #1 (Remarks to the Author):

I am satisfied with the revisions. This is an important study in a rapidly developing area.

Author Response to Reviewers

REVIEWERS' COMMENTS

Reviewer #1 (Remarks to the Author):

I am satisfied with the revisions. This is an important study in a rapidly developing area.

The authors thank the Reviewer for their constructive review of our study.